# MAPPING LANGUAGE MODELS TO GROUNDED CONCEPTUAL SPACES

**Roma Patel & Ellie Pavlick**
Department of Computer Science
Brown University
{romapatel,ellie_pavlick}@brown.edu

## ABSTRACT

A fundamental criticism of text-only language models (LMs) is their lack of *grounding*—that is, the ability to tie a word for which they have learned a representation to its referent in the non-linguistic world. However, despite this limitation, large pre-trained LMs have been shown to have a remarkable grasp of the conceptual structure of language, as demonstrated by their ability to answer questions, generate fluent text, or make inferences about entities, objects, and properties that they have never physically observed. In this work we investigate the extent to which the rich conceptual structure that LMs learn indeed reflects the conceptual structure of the non-linguistic world—which is something that LMs have never observed. We do this by testing whether the LMs can learn to map an entire conceptual domain (e.g., direction or colour) onto a grounded world representation given only a small number of examples. For example, we show a model what the word *"left"* means using a textual depiction of a grid world, and assess how well it can generalise to related concepts, for example, the word *"right"*, in a similar grid world. We investigate a range of generative language models of varying sizes (including GPT-2 and GPT-3), and see that although the smaller models struggle to perform this mapping, the largest model can not only learn to ground the concepts that it is explicitly taught, but appears to generalise to several instances of unseen concepts as well. Our results suggest an alternative means of building grounded language models: rather than learning grounded representations "from scratch", it is possible that large text-only models learn a sufficiently rich conceptual structure that could allow them to be grounded in a data-efficient way.

## 1 INTRODUCTION

Large pre-trained language models (LMs) trained on text corpora have shown remarkable progress on a range of natural language understanding tasks (Radford et al., 2019; Brown et al., 2020). Such models have demonstrated their ability to generate fluent dialogue (Brown et al., 2020), make commonsense inferences Zellers et al. (2019), and reconstruct taxonomies and word relations (Chen et al., 2020). However, it has been argued that true meaning cannot be learned from the *form* of language alone (i.e., from text) because it is a word's *use* in the non-linguistic world that imparts it meaning (Bender & Koller, 2020; Bisk et al., 2020). For example, although LMs might learn from textual co-occurrences that the words *north* and *south* are opposites, the grounded meaning of these words, i.e., the direction that you should travel if you are told to go *north*, is something to which these models, by definition, do not have access during training.

While it is indisputable that text-only models do not learn representations of concepts that are grounded in the non-text world, it is possible for the structure of relations between concepts in text form to be identical to what a grounded model would learn. In principle, it is therefore possible for a text-only model's conceptual space to be *isomorphic* to the "true" (i.e., grounded) conceptual space (Merrill et al., 2021). In this work, we investigate whether this is the case by asking whether models can learn to ground an entire domain (e.g., direction) after grounding only a subset of the points in that domain (e.g., *left*). Specifically, for generative LMs that have been trained only on large text corpora, we "orient" the models by showing them how some word forms (that they have learned during training) are used in simple text worlds—for example, what the direction *north* maps

to in a textual representation of a grid world (see Figure 1). We then evaluate two types of generalisation. First (§3.1), we evaluate generalisation to unseen worlds. For example, if the model has seen several realisations of the word *north* in different grid worlds, can it correctly identify *north* in an unseen world (e.g., one of a different size or shape)? Second (§3.2), we evaluate generalisation to unseen but related concepts. For example, if the model has been shown grounded representations of *north* and *east*, can it correctly identify *south* and *west*, even though it was never shown them? We find that although the small language models (GPT-2 models that contain on the order of 100M parameters) cannot perform either generalisation well, the largest model (a GPT-3 model containing 175B parameters) can indeed learn groundings in the conceptual worlds we build. We analyse the predictions and errors made by models (§3.3) and find that the errors made by the small models are often due to a failure to recognize the domain, and thus generating random words by default. In contrast, the errors made by the largest model are often intuitive, e.g., predicting in-domain concepts that are reasonable substitutes for the target (e.g., *maroon* versus *dark red*).

## 2 EXPERIMENTAL DESIGN

### 2.1 MODELS

We test five autoregressive Transformer language models (Vaswani et al., 2017) of varying size, specifically the GPT-2 (Radford et al., 2019) and GPT-3 (Brown et al., 2020) models. Our smallest model contains 124M parameters, and the others follow increasing model sizes (355M, 774M, 1.5B and 175B parameters). All models are pre-trained on differently filtered versions of the OpenAI Web-Text dataset Radford et al. (2019), composed of 40GB of English web text available on the internet. We generate up to 5 tokens per prompt and, to improve the robustness of our analyses, generate 3 samples per prompt. We use a temperature of 1 during generation and sample from the softmax probabilities produced at each time step using nucleus sampling (Holtzman et al., 2019) with p=0.85. We include more detail on the models and their training data in Appendix A.

### 2.2 IN-CONTEXT LEARNING

Several studies (Brown et al., 2020; Reynolds & McDonell, 2021) have shown that instead of fine-tuning generative LMs–i.e., a process that updates parameters learned by the model during pre-training–it is possible to achieve competitive performance by giving the model a small number of training examples within the prompt. This is often referred to as "in-context learning" or "few-shot prompting". Specifically, a prompt includes $n$ task examples that include a question prefix (e.g., **"World:"**) followed by the question, and an answer prefix (e.g., **"Answer:"**) followed by the answer to the question. After giving the model $n$ examples in this manner, the prompt ends with a new question and only an answer prefix after which the model is expected to generate an answer to the last question, following the prompt format it has seen. By enumerating over all questions in the test set, we can obtain a model-generated answer for every test set question that we wish to evaluate. There are no gradient updates to any model parameters using this approach.

### 2.3 GROUNDED CONCEPT DOMAINS

The models that we test, by construction, can receive only text inputs. Thus, we focus on a set of grounded domains for which it is possible to faithfully represent the grounded meaning in text form. We briefly describe these domains below, summarise them in Figure 1, and describe in detail, the prompt creation process for each generalisation task in Sections §3.2 and §3.1. We discuss the possibility of expanding this set of concepts in future work in Section §4.

**Spatial Terms** We consider 6 spatial concepts: *left, right, up, down, top, bottom*. Each of the above concepts can be represented in a grid world using the position of a special character (here, a '1') in the world. To do this, we create grid world environments of varying sizes (where the number of rows and columns ranges from 1 to 8), where each world consists of '0's and a single '1'.

**Cardinal Directions** We consider eight cardinal directions: *north, south, east, west, northeast, northwest, southeast, southwest*. These are similar to spatial terms, except that they include compo-

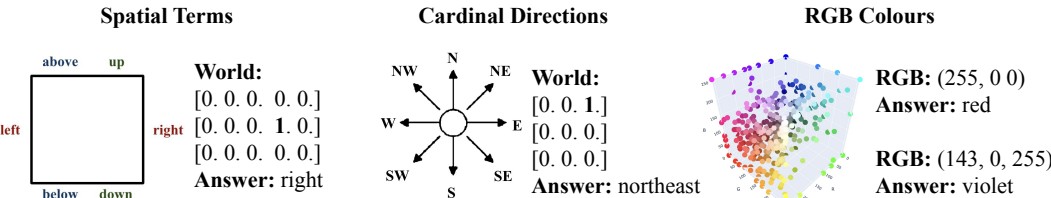

Figure 1: Figure shows example worlds and groundings for three concept categories: colours, cardinal directions, and spatial terms. For each of the three domains, the left figure in each shows the full set of grounded concepts in the domain. To the right, we see example world representations with textual instantiations of the groundings that serve as prompts for language models.

sitional terms (e.g., *northeast*). We use grid-worlds of the same format as spatial terms to represent these concepts.

**Colour Terms**   We consider colour terms in a three-dimensional space, using a dataset of 367 RGB colours (Abdou et al., 2021) that contains colour names (e.g., *red, cyan, forest green*) each associated with an RGB code (e.g., $(255, 0, 0)$). Therefore, the world representation in this case is not a grid-world, but an RGB code associated with every colour name. Figure 1 shows example RGB codes and colours that serve as part of a prompt given to the models we test.

## 2.4   ROTATED WORLDS TO CONTROL FOR MEMORISATION

**Motivation**   The GPT-x models that we use have been trained on the CommonCrawl corpus (Radford et al. (2019)), a collection of documents that contains web text freely available on the internet. Since we provide instantiations of grounded concepts in text form, it is very plausible that the domains described above have been encountered verbatim during training. For example, for spatial terms such as *left*, a model might have seen instances of matrices and linear algebra terms with the word *left* in close proximity; for colour terms, tables that map RGB colour codes to colour names are pervasive in web-text. We therefore include a control task in our experimental setup such that the model cannot succeed using simple memorisation. Rather, success on a task requires the model to truly perform a conceptual mapping between the ungrounded and grounded representations.

**Isomorphism as a Control**   We use the concept of *isomorphism* to control for memorisation. Intuitively, imagine a situation where you are lost in the woods. Once pointed in the direction of north, you instantly know which way is south. However, this ability is not dependent on having been correctly pointed north–if someone were to incorrectly point east and tell you this was north, you would readily infer west to be south. This is because your reasoning depends on your knowledge of the relation between north and south, and between north and the world, rather than on having memorized the "true" grounding of each concept independently.

By the same logic, if a model is learning a grounding function, this should be dependent on the world in which it is being grounded. For example, in different worlds where the word *red* grounds to different points in space, the word *blue*, by analogy, shares a fixed conceptual relation to the word *red*. Therefore, it should ground in correspondingly equidistant ways in the two worlds. A model's ability to learn two different grounding functions $f$ vs. $g$, should not be dependent on what the actual points ground to, as long as the structural relations between concepts in the space are preserved. Further, this should hold for all such isomorphic transformations that preserve the structure of the space, and importantly, it should *not* hold for random perturbations that distort the structure of the space, since such distortions would break the assumption that the relation between *red* and *blue* in ungrounded space is analogous to that in grounded space.

**Implementation**   In the colour domain, since colour concepts exist in a 3D world of RGB codes, we rotate each point around a fixed axis by a certain degree to create a new isomorphic world. We repeat this control three times (for $90°$, $180°$ and $270°$ rotations) and average over the rotations in our evaluations. For cardinal directions, we rotate each cardinal concept by $90°$ in two dimensions, and do this three times and average over rotations. Since the spatial terms exist as pairs, we simply

swap the groundings of alternate terms (e.g., *left* and *right*). For random worlds that do not preserve the structure between word forms, we randomly assign a concept name (e.g., *red*) to a point in the world, and we do this for all concept names and categories to obtain a random world for each. Figure 2 shows example transformations of colours on rotating by $90°$, as well as randomly rotating points.

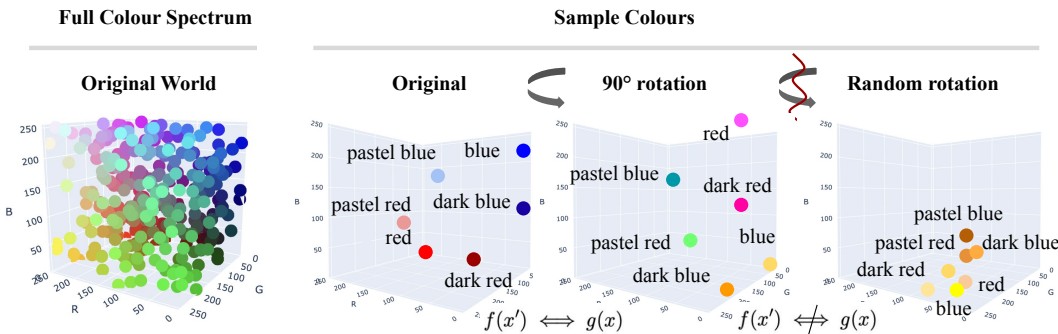

Figure 2: Figure shows how colours and modifiers transform in rotated worlds. The leftmost figure shows a full 3-D colour space of 367 colours. The three figures on the right, show four sample colours in their original world, a world rotated by $90°$, and a randomly rotated world, showing how the structure of the space is preserved or distorted in isomorphic or random rotations respectively.

## 2.5 EVALUATIONS

**Experimental Logic**   We report model performance in three settings: the original ("true") world (e.g., that in which *red* maps to the actual RGB code for red), an average over three rotated worlds (e.g., worlds in which *red* maps to some other RGB code, but relations between colors are preserved), and a random world (in which relations between mappings are not preserved). If a model is performing the conceptual mapping in the desired way, we expect that performance should be high in the true world and that there should be no significant degradation in performance when moving to rotated worlds. We also expect that performance should be low in the random world.

**Metrics**   When given a prompt, a generative LM is free to generate any number of tokens until having generated the EOS token that halts further generation. Since classification tasks usually correspond to labels that contain only a few words, the standard approach is to let the model generate an entire sequence of text and to then cut off the generation to the first $n$ tokens (where typically $n < 10$). From the prompting mechanism, the model should learn to follow the prompt format to generate one label (e.g., instead of every label in succession), and since it does not receive any gradient updates during this learning, there is no incentive for it to predict all related labels within a $n$-token generation. We see that this is true in practice and show example generations in Appendix E. We set $n = 5$ and then report the following metrics.

TOP-1 ACCURACY   If the ground-truth answer or any substring thereof lies in the generated answer (i.e., the first $n$ tokens of the full generation), the model gets perfect accuracy. If the ground-truth answer does not exist in the generated answer, or exists in the generation outside of the cutoff limit, the model gets a 0 accuracy. For example, for the ground truth answer *deep tuscan red*, if the model generated answer is *tuscan red* or *red* the model gets a perfect accuracy, but if the model generated answer is *deep red* or *wine* or *vermilion*, the model gets an accuracy of 0. We also compute the same metric using exact match (i.e., the model is only correct if it generates exactly *deep tuscan red*). We find the values are lower but the trends are the same; see Appendix **??**..

TOP-3 ACCURACY   This metric is analogous to Top-1 except that instead of only considering the most probable generation from the model, we collect the second and third most probable answer sequences as well. The model gets a perfect accuracy if the correct answer exists in any of the three generated answers. If the correct answer does not exist in any of the 3 generated answers, or exists in any of the generations outside of the cutoff limit, the model gets an accuracy of 0. Again, see Appendix **??** for results using exact match.

**Example Input (20 in-context-learning examples followed by prompt)**       **Example Model Outputs**

```
World:           World:           World:           World:
[0. 0. 0.]       [0. 0. 0.]       [0. 0. 0.]       [1. 0.]
[0. 0. 0.]       [0. 0. 0.]       [0. 0. 0.]       [0. 0.]
[0. 0. 1.]       [0. 0. 1.]       [0. 0. 1.]       [0. 0.]
Answer: right    Answer: right    [0. 0. 0.]       [0. 0.]
                                  Answer: right    Answer: left
World:           World:
[1. 0. 0. 0.]    [0. 0.]                           World:
Answer: left     [1. 0.]          World:           [1. 0. 0. 0.]
                 [0. 0.]          [0. 1. 0. 0.]    [0. 0. 0. 0.]
...13 more...    Answer: left     Answer: left     Answer:
```

| GPT-2 (124M) | |
| --- | --- |
| world | P=0.09 |
| 0. 0.]] | P=0.08 |
| [0 [0 | P=0.01 |

| GPT-3 (175B) | |
| --- | --- |
| left | P=0.20 |
| right | P=0.11 |
| leftmost | P=0.01 |

Figure 3: Figure shows example worlds and groundings for the spatial domain. The left panel shows example grid worlds where the location of the 1 denotes a point grounded in the world, with a corresponding text instantiation of that particular concept. The two panels on the right show example outputs from the smallest and largest models. We show the top three most probable words from each model along with the probability of that word. When computing Top-1 accuracy, we only consider the first, however, we consider all 3 when computing Top-3 accuracy.

GROUNDING DISTANCE    For analysis purposes (§3.3), we wish to assess how far off models are in cases where they are wrong. For this, we need to quantify the distance between the model's predicted answer and the ground truth answer. For example, in the colour domain, the distance between two points can be computed as the Euclidean distance between two RGB codes. For every answer generated by the model (e.g., the word *pink* in the colour domain), if the answer is an acceptable grounded term that exists in the world, we can compute its distance to the true grounding (e.g., the colour *red*). However, if the generated answer was an unrelated word (e.g., the word *cat*) that does not exist in the same domain, no distance can be computed in the world. Therefore, we calculate a distance metric as the Euclidean distance between two points in space when the generated answer falls in the domain. When the generated answer does not fall in the domain, we set the distance to a number significantly higher than the largest distance between two in-domain concepts. We provide equations and details on calculation of this metric in Appendix C.

**Baselines**    Given that the language models are free to generate any word that exists in their vocabulary as a potential answer, we choose two random baselines over vocabulary words against which to compare model performance. We use **R-IV** (i.e., random in-vocabulary) to denote a baseline that randomly selects from among all words in the model's vocabulary. We use **R-ID** (i.e., random in-domain) to denote a baseline that randomly selects from amongst only the in-domain words (e.g., from colour terms); this is 6, 8 and 367 words respectively for the spatial, cardinal and colour categories. Note that the generative LMs do not have such a domain restriction over words, as they are free to choose any token from the full vocabulary (like R-IV).

## 3 RESULTS

### 3.1 GENERALISATION TO UNSEEN WORLDS

Our first investigation looks into how well models generalise known concepts to unseen worlds. For example, if a model has seen a few examples of a concept (such as *left*) depicted in some grid worlds, can it correctly identify an instance of t*left* in a different grid world (e.g., one with a different size or orientation)? Note that we can only conduct this type of evaluation in the spatial and cardinal domains, since, for the colour domain, there is only ever one world, with one grounding for each concept (e.g., the colour *red* has exactly one grounding to a 3-digit RGB code in a 3-D RGB space).

**Data**    We create prompts that include 20 examples of grounded concepts in a set of grid worlds. For each domain (e.g., cardinal directions that contain 8 concepts, and spatial terms that contain 3 pairs of 2 concepts), we include a (roughly) equal sample of concepts among these 20 examples. Then, we append a held-out grid world to the end of the prompt and evaluate whether or not the models generate the correct concept label for these unseen worlds. Figure 3 shows example prompts

| | Top-1 Accuracy | | | | | | Top-3 Accuracy | | | | | |
|---|---|---|---|---|---|---|---|---|---|---|---|---|
| | Spatial | | | Cardinal | | | Spatial | | | Cardinal | | |
| | Orig. | Rot. | Rand. | Orig. | Rot. | Rand. | Orig. | Rot. | Rand. | Orig. | Rot. | Rand. |
| **R-IV** | 0.00 | 0.00 | 0.00 | 0.00 | 0.00 | 0.00 | 0.00 | 0.00 | 0.00 | 0.00 | 0.00 | 0.00 |
| **R-ID** | 0.16 | 0.16 | 0.16 | 0.13 | 0.13 | 0.13 | 0.16 | 0.16 | 0.16 | 0.13 | 0.13 | 0.13 |
| **124 M** | 0.11 | 0.10 | 0.10 | 0.13 | 0.12 | 0.11 | 0.23 | 0.21 | 0.10 | 0.25 | 0.24 | 0.10 |
| **355 M** | 0.12 | 0.12 | 0.10 | 0.11 | 0.14 | 0.10 | 0.24 | 0.25 | 0.15 | 0.23 | 0.14 | 0.12 |
| **774 M** | 0.08 | 0.09 | 0.10 | 0.11 | 0.12 | 0.11 | 0.12 | 0.19 | 0.14 | 0.18 | 0.17 | 0.11 |
| **1.7 B** | 0.10 | 0.11 | 0.11 | 0.10 | 0.11 | 0.10 | 0.11 | 0.18 | 0.15 | 0.12 | 0.12 | 0.13 |
| **175 B** | **0.45** | **0.44** | **0.16** | **0.43** | **0.46** | **0.18** | **0.76** | **0.75** | **0.19** | **0.88** | **0.76** | **0.21** |

Table 1: Table shows evaluations for generalisation to unseen worlds. Rows show model sizes for GPT-2 models (124M to 1.5B) and the 175B GPT-3 model. Columns show metrics for the original, rotated and random worlds for each of the concept categories. We report both Top-1 and Top-3 accuracy in the first and second columns of each rotated world.

given to the model and example generations from three different models. We report results averaged over all generations in Table 1. In general, we see the desired trend in which model performance on the original and rotated worlds is well above that on the random world. We do not see consistent or significant performance degradation when moving from the original to the rotated world, suggesting performance is not due to simple memorisation. Comparing across models, we see that the smaller models struggle to even learn concepts that were taught to them in a few-shot manner. For example, for the spatial category, the performance of the smallest model is below that of a baseline which guesses randomly among in-domain words, suggesting the model even fails to learn the general domain of the task (discussed more in Section 3.3). In contrast, the largest model (GPT-3) has a 45% Top-1 accuracy and a 76% Top-3 accuracy for the spatial category.

## 3.2 GENERALIZATION TO UNSEEN CONCEPTS

Our primary interest here is in the model's ability to map its ungrounded conceptual structure to a grounded world. Specifically, we want to see that the model is able to ground an entire conceptual domain when only taught how to ground a small subset of the points in that domain. To test this, we show models example concepts within a sub-space of the domain (e.g., *north, east*), while holding out other concepts (e.g., *south, west*). We then test them on instances of held-out concepts. Figure 6 depicts this setup in the colour domain: we train the model using primarily shades of red, but test on shades of blue.

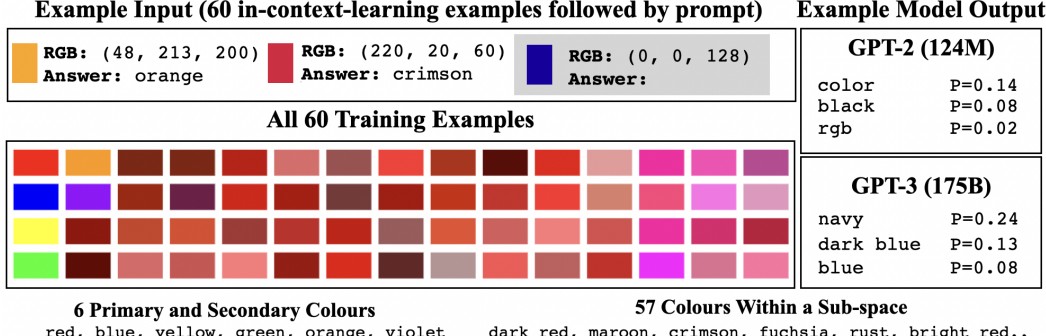

Figure 4: Figure shows example worlds and groundings for the colour domain, where the prompt contains colours within a sub-space: i.e., training on primary and secondary colors plus shades of red, but then testing on navy blue. The left panel shows the full set of training examples the model sees. The right shows example outputs from the smallest and largest models, with the top three most probable words from each model along with the probability of that word.

| | | Spatial | | | Cardinal | | | Colours | | |
|---|---|---|---|---|---|---|---|---|---|---|
| | | Original | Rotated | Random | Original | Rotated | Random | Original | Rotated | Random |
| | **R-IV** | 0.00 | 0.00 | 0.00 | 0.00 | 0.00 | 0.00 | 0.00 | 0.00 | 0.00 |
| | **R-ID** | 0.16 | 0.16 | **0.16** | 0.13 | 0.13 | 0.13 | 0.00 | 0.00 | 0.00 |
| | **124 M** | 0.10 | 0.11 | 0.04 | 0.11 | 0.10 | 0.05 | 0.08 | 0.09 | 0.03 |
| **Top-1** | **355 M** | 0.10 | 0.10 | 0.04 | 0.10 | 0.11 | 0.06 | 0.06 | 0.07 | 0.04 |
| **Accuracy** | **774 M** | 0.09 | 0.11 | 0.03 | 0.13 | 0.12 | 0.08 | 0.11 | 0.09 | 0.01 |
| | **1.5 B** | 0.14 | 0.14 | 0.12 | 0.13 | 0.14 | 0.10 | 0.10 | 0.09 | 0.06 |
| | **175 B** | **0.28** | **0.27** | 0.13 | **0.30** | **0.29** | **0.08** | **0.23** | **0.21** | **0.11** |
| | **R-IV** | 0.00 | 0.00 | 0.00 | 0.00 | 0.00 | 0.00 | 0.00 | 0.00 | 0.00 |
| | **R-ID** | 0.16 | 0.16 | 0.16 | 0.13 | 0.13 | 0.13 | 0.00 | 0.00 | 0.00 |
| | **124 M** | 0.13 | 0.12 | 0.07 | 0.09 | 0.09 | 0.08 | 0.06 | 0.05 | 0.04 |
| **Top-3** | **355 M** | 0.24 | 0.17 | 0.15 | 0.19 | 0.17 | 0.10 | 0.14 | 0.11 | 0.12 |
| **Accuracy** | **774 M** | 0.19 | 0.24 | 0.12 | 0.17 | 0.15 | 0.11 | 0.15 | 0.16 | 0.14 |
| | **1.5 B** | 0.32 | 0.29 | 0.20 | 0.21 | 0.20 | 0.14 | 0.19 | 0.18 | 0.16 |
| | **175 B** | **0.64** | **0.65** | **0.21** | **0.60** | **0.61** | **0.09** | **0.34** | **0.36** | **0.13** |

Table 2: Table shows evaluations for generalisation to unseen concepts for the sub-space split of training/test data. Rows show model sizes for GPT-2 models (124M to 1.5B) and the 175B GPT-3 model. Columns show metrics for the original, rotated and random worlds for each of the concept categories. We report both Top-1 and Top-3 accuracy. R-IV refers to a random baseline that randomly selects a word out of the total vocabulary, R-ID refers to a baseline that randomly selects a word out of the set of words in the concept category.

**Data** To create sub-spaces for the colour domain, we create prompts that contain 3 primary, 3 secondary colours, and 57 other colours within a sub-space, determined in the following way. For a certain colour (e.g., *red*) we consider a *sub-space* in the world to be a space of colours that lies within a Euclidean distance of 150 from that colour. We create 6 such sub-spaces, centered around each of the primary and secondary colours, and report generalisation results averaged over the 6 sub-spaces. Figure 4 shows the sample of colours within a sub-space centered around the colour red, that serve as training samples within the prompt. For the cardinal directions, we show models examples of concepts in one sub-space of the world (e.g., *north, east, northeast*) and then test them on concepts in an entirely different sub-space (e.g., *south, west, southwest*). The training split therefore contains worlds annotated with one sub-space, and we test on the remaining held-out concepts. We do this for all sub-spaces and average over them.

We report results on all concept categories in Table 2. As before, we see that models do not appear to be exploiting simple memorisation (evidenced by similar performance in original vs. rotated worlds) and that only the largest models appear capable of getting reasonable performance on the task. That said, the largest GPT-3 model achieves impressive results given the difficulty of the task. For example, it achieves over 40% Top-1 accuracy in the color domain, which requires generating a label like *violet* despite having never seen such a label during training (Figure 4).

### 3.3 ERROR ANALYSIS

Given the results above, we investigate "how wrong" models are when they fail to produce the expected ground-truth label. One clear trend of the large model is the ability to generate "on topic" (i.e., in-domain) responses, regardless of whether or not those responses are correct. For example, if the expected response to an input is *"left"*, a model which produces *right* is wrong in a very different way than a model that produces nonsensical outputs such as *"[0[0"* or function words such as *"the"* (see Figure 3). We see that the largest 175B parameter model almost always produces in-domain answers. We evaluate this by checking whether the generation lies in the set of related words for that category (e.g., all colours, spatial, or cardinal words, respectively). We see that the smaller models fail to do this. That is, their generations tend to be unrelated words that might have had high prominence (e.g., function words). On evaluating accuracy of generations being "in-domain",

we see that the smallest model has only a $53\%$ accuracy while the largest has a $98\%$ accuracy of generating in-domain answers.

Second, we evaluate grounded distance (§2.5) to measure the degree of correctness of responses. In the colour domain, the colours *dark red* and *wine* are close enough in space that they might be intuitive alternate answers for one another. However, our top-1 and top-3 metrics only assess string matches and do not account for this. Thus, we look at the distance between the colour denoted by the predicted label (when it exists, i.e., when the model generated a legitimate colour name) and the colour the model was asked to label. The lower this distance is, the "less wrong" the model's prediction is.

Table 3 reports evaluations measured by grounding distance for the colour domain. For every test instance, we compute the distance between the model predicted grounding and the true grounding as defined in Equation C.3. We then average over all computed distances to report one number that tells us how close, on average, model predictions are to the true grounding in the world. We show example visualisations of colours in RGB space that lie within a certain distance threshold of each other. We see, for the largest model, the distances between model-predicted groundings from the true grounding are significantly lower than random. Such a result suggests that the model's errors are often justifiable and the scores given by our string-matching metrics might be an underestimate of the model's true performance.

|  | 124M | 355M | 774M | 1.5B | 175B |
|---|---|---|---|---|---|
| $C$ | 328.3 | 309.5 | 209.6 | 190.7 | 96.3 |
| R-IV | 334.9 | 334.9 | 334.9 | 334.9 | 334.9 |
| R-ID | 174.9 | 174.9 | 174.9 | 174.9 | 174.9 |

Table 3: Table shows average distance (lower is better) between model-predicted groundings and true groundings in the world, averaged over all instances in the test set. We see that the largest model has an average distance of predictions significantly lower than random

| True $G$ | Predicted $G$ & Distance |
|---|---|
| dark red | wine (76.5), light crimson (208.1) dark slate gray (144.7) |
| light green | beige (126.6), light sea green (129.7) cerulean (185.7), violet (262.6) |

Table 4: Table shows example model predictions (from the GPT-3 model) and distances from the true groundings in RGB space. The first column shows the true concept while the second column shows model predictions and their distances from the true concept.

## 4    DISCUSSION

Our empirical results suggest that very large LMs (specifically, GPT-3), even when trained only on text, learn a conceptual space that can, at least in some settings, be "grounded" using only a small number of training examples. The fact that these models succeed even in isomorphic rotated worlds suggests that these models are not succeeding via naive memorisation. Rather, this suggests that they may be exploiting something about the conceptual structure of the space learned from text in order to map onto a new space that was not explicitly encountered during training.

A major limitation of our approach is that there are some grounded concepts (e.g., visual and sensory inputs) that cannot be easily encoded in text form. By construction, the LMs that we use are restricted to text-only inputs, thus our focus is on domains (e.g., colours and directions) that have a well-defined textual representation. This is only a small set of all the potential grounded concepts we would wish to teach LMs. Although many forms of data can be coerced into text format (e.g., we represent color using discrete digits to represent RGB space), complex concepts may loose fundamental aspects of their meaning when represented in this way. For example, for color, a coarse-grained notion of numeric proximity, derivable from text (Wallace et al., 2019; Naik et al., 2019), may be sufficient to differentiate the concepts we explore, but for more complex visual inputs (e.g., the output of a CNN image encoder), a text-based numeric representation is unlikely to capture the necessary degree of nuance. Future work would need to consider ways of adapting GPT-3-like models to accept non-textual inputs while still exploiting the text-based conceptual structure.

If such limitations were addressed, our results are suggestive of a potentially promising way in which text-only training could support general purpose, grounded models of meaning. Specifically, our results imply that the conceptual space a model learns from text might be nearly isomorphic to

what it would learn from interacting in a grounded world, and that models can be taught to map between those conceptual spaces without requiring explicit grounding for every concept. This is exciting, as domain-general text corpora are readily available, while domain general multimodal corpora–e.g., containing sufficient information on abstract concepts such as emotions (*happy*) or time (*used to*)–might be difficult or impossible to collect. If models like GPT-3 could be adapted to receive non-text prompts (as discussed above), our results suggest that the rich conceptual structure such models learn from text could be bootstrapped into powerful grounded models of language.

## 5    RELATED WORK

There is a significant amount of work that focuses on understanding how LMs represent and reason about concepts, as well as work that directly attempts to build models that take text inputs and ground them to elements in the world. We situate our work within these two bodies of literature: one that investigates how LMs understand linguistic phenomena and word meaning, and another, that attempts to situate language in models of the world. We describe each body of work below.

**Meaning and Understanding in LMs**    With the advent of large LMs of increasing orders of magnitude, there has been speculation on the capabilities of such models, and whether they truly understand the meaning of the words they are learning representations for. Several works that attempt to probe linguistic phenomena in LMs, show that the representations learned by such models encode syntactic dependencies and coreference information (Tenney et al., 2019) and word-sense information (Chen et al., 2020). Work that investigates the ability of models to form word associations (Hwang et al., 2020), finds that large LMs can indeed perform such a task; suggesting that pretrained LMs not only recognize that entities are related, but can differentiate *how* they are related. Especially relevant to our work is recent work that investigates alignment of language models to colours (Abdou et al., 2021) or state changes (Li et al., 2021). Our work is complementary to this prior work, and we specifically ask whether the text space can be reliably mapped onto the grounded space.

**Natural Language Grounding**    There is an increasing amount of work, usually at the intersection of NLP and fields like vision and reinforcement learning, that aims to use natural language to instruct agents about aspects of the world. In the case of vision, this could be to learn correspondences between language descriptions and pixels in an image (Eichenberg et al., 2021; Tsimpoukelli et al., 2021), or in the case of RL, to build agents that understand natural language instructions in order to take actions in a world that follow the instruction—for example to navigate to a goal (Artzi & Zettlemoyer, 2013; Patel et al.), or to solve tasks in different languages (Ku et al., 2020). Most of these tasks focus on training LMs from scratch, however usually with inputs that contain both textual information as well as grounded world information. Our work is different in that we attempt to take an LM that was previously trained only on text, and attempt to teach it a concept in the world without re-training it. With only a few samples of what the concept grounds to, we investigate how well large LMs can use the structure of language and associations between word forms in order to generalise to grounded concepts.

## 6    CONCLUSION

This work investigates the extent to which large language models, trained only on text can be taught to map previously learned word forms onto conceptual worlds. We investigate several generative language models in colour and direction domains, represented as discrete grid worlds given to the model as text input. With only a few examples of such grounded instances, although smaller models struggle to generalise from text to grounded concepts, we see that the largest model does indeed learn the space of concepts that we test it on. We analyse where models fail and see that the smallest models often produce random, unrelated outputs, but the errors made by the largest model are quite intuitive, for example, predicting colours very close in space to the true grounding. We discuss the limitations of focusing on text-only inputs to teach models grounded concepts, as well as the implications of such work for allowing large language models to be mapped, in a data-efficient way, to the grounded world.

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

## APPENDIX

We provide, as supplementary material, additional information about the data generation, models used, examples of model generations, as well as additional results across all models.

## A    MODELLING DETAILS

We use a GPT-3 model Brown et al. (2020) and four GPT-2 Radford et al. (2019) models from the Hugging Face Transformer Wolf et al. (2019) library. Each of these is a pretrained autoregressive transformer model, trained on the OpenAI WebText corpus, containing around 8 million documents. The top 15 domains by volume in WebText are: Google, Archive, Blogspot, GitHub, NYTimes, Wordpress, Washington Post, Wikia, BBC, The Guardian, eBay, Pastebin, CNN, Yahoo!, and the Huffington Post. Individual model parameters and layers are shown in Table 5. The pretrained models use byte-pair encoding (BPE) tokens Sennrich et al. (2015) to represent frequent symbol sequences in the text, and this tokenisation is performed on all new input prompts to generate text from the model. We report the hyperparameters used by the pretrained model in Table 6.

| Parameters | Layers |
|---|---|
| 124M | 12 |
| 355M | 24 |
| 774M | 36 |
| 1.5B | 48 |
| 175B | 96 |

Table 5: Table shows model architecture details for the GPT-3 and four GPT-2 models we use.

| Hyperparameter | Selection |
|---|---|
| number of samples | 3 |
| nucleas sampling $p$ | 0.85 |
| temperature | 1 |
| max length | 5 |

Table 6: Table shows model architecture details for the models we use.

# B   DATA GENERATION

In this section we describe how we create prompt data for the concept categories we wish to evaluate. We describe what the "worlds" look like for each category, as well as detail on the generation of such worlds and division into train and test splits, for colours in Section B.1 and for both spatial and cardinal concepts in Section B.2.

## B.1   COLOUR CONCEPTS

We draw from an existing dataset containing RGB codes associated with colour names, for 367 colours. In this concept category, a grounded realisation of a colour in the world, is simply its RGB code i.e., the point that it grounds to in 3-dimensional colour space. Therefore, all grounded examples in prompts follow the format of **RGB:** (x, y, z) followed by **Colour:** concept name , where the items in red are replaced with instances of RGB codes and names from the dataset of colours we useAbdou et al. (2021). This gives us a total of 367 samples of RGB codes paired with colour names. We create training and testing splits for different generalisation evaluations in the following way.

### B.1.1   GENERALISATION TO UNSEEN CONCEPTS

**Random Split**   We create prompts to perform this experiment in two ways. The first, as reported in §3.2, creates a "train split" i.e., samples within the prompt, by first selecting the 3 primary and 3 secondary colours to always be in the prompt. We then sample 64 other colours from the set of colours to be part of the train split. The prompt therefore contains 70 samples of the question and answer prefixes followed the RGB codes and colour names. For every sample in the test set, we create a new prompt that appends a question prefix and the RGB code of that sample to the end of the prompt, followed by the answer prefix (with no answer). For each prompt, the model is then required to generate an answer. Figure 6 shows the random sample of colours in the prompt and we report results on these in Appendix Table F.1.

**Sub-space Split**   We then perform another experiment to evaluate a models ability to learn concepts within a sub-space, and generalise to a whole new sub-space. Here, the prompt contains 3 primary and 3 secondary colours, as well as 57 other colours that we select in the following way. For each of the 6 primary and secondary colours, we consider a *sub-space* in the world to be a space of colours that lies within a Euclidean distance of 150 from that colour.[1] Since the model has seen the 6 primary and secondary colours, as well as a certain space of the colour world, we wish to evaluate how well the model can generalise to a new sub-space of the world, by using the colour

---

[1]We show visualisations of such colour sub-spaces in the main paper in Figure 4

word-form associations. We then report generalisation results averaged over the 6 sub-spaces. Figure 4 shows the sample of colours within a sub-space centered around the colour red, that serve as training samples within the prompt. We report results on these in the main paper in Table 2.

## B.2 SPATIAL AND CARDINAL CONCEPTS

We consider a "world" to be a 2D matrix filled with 0s and containing exactly one 1, where the position of the 1 (i.e., the determining object) refers to the concept of interest (for e.g., a spatial location like *"left"* or a cardinal direction like *"north"*. We create grid worlds filled with 0s and one 1, of row and column ranges from $(1, 8)$ giving us 672 total grid worlds.[2] For each grid world, all grounded examples follow the format of **World:** [1. 0. 0. 0.] followed by **Direction:** concept name . where the items in red are replaced with instances of different grid worlds and corresponding concepts, based on the location of the 1. We create training and testing splits for different generalisation evaluations in the following way.

### B.2.1 GENERALISATION TO UNSEEN WORLDS

Our primary investigation here is to assess whether the models can generalise to new worlds that differ in size, or location of the determining object. Each of the 672 worlds we create differs from one another in either aspect i.e., either the size of the world is different, or, when worlds are of the same size, the location of the 1 is in a different position. Therefore, we randomly sample 20 worlds that contain instances of concepts to serve as part of the prompt. The prompt therefore contains a question prefix followed by the grid world on a new line, followed by the answer prefix and the concept name. We ensure that there is a roughly equal distribution of concepts in the train split (e.g., if there are 8 concepts split over 20 samples, we ensure that 2 of each concept exist in the prompt, and then fill the remainder of the prompt by randomly sampling from the concepts). We then create a new prompt for every sample in the test set by appending the world representation and answer prefix to the end. We report results on this in the main paper in Table 1.

### B.2.2 GENERALISATION TO UNSEEN CONCEPTS

Since we wish to assess a model's ability to generalise to new concepts, here, we hold out concepts in the test set. Specifically, for every domain contain $n$ concepts (e.g., 8 for cardinal directions), the train split contains an equal distribution of $n - 1$ concepts. We then test on samples that contain the last held-out concept, as well as the seen concepts, to assess how well the model has learned seen concepts, and generalises to unseen concepts. We report results on this in Appendix Table 14, and Figure 7 shows an example split of data that is held-out during test time. We note that this is a random split, unlike the sub-space split that specifically holds out a set of colours based on Euclidean distance.

### B.2.3 GENERALISATION TO SUB-SPACES

Similar to the colours, for the cardinal directions, we also report in the main paper, results on a sub-space generalisation task. Specifically, we show models examples of concepts in one sub-space of the world (e.g., *north, east, northeast*) and then test them on concepts in a different sub-space (e.g., *south, west, southwest*). We do this for all sub-spaces and average over them. As seen in Table 11, we see that the largest model can perform this task to some degree, achieving a $60\%$ accuracy.

### B.2.4 GENERALISATION TO COMPOSITIONS

Here, we specifically test model performance when holding out any instance of compositions of directions. Therefore, the training split contains examples of worlds annotated with the concepts *north, south, east, west*, and the test split contains examples of compositions of concepts i.e., *northeast, southeast, northwest, southwest*. We report results in Table 10, and we see that here, even the largest model fails to perform the task. This means that when models have never seen any instance of a composition, they do not perform these compositions themselves, however, as seen in Table

---

[2]We exclude grid worlds where the position of the 1 is in an ambiguous position for that concept category e.g., directly in the center for a cardinal task, or in the middle row for a spatial task.

10, and earlier results, when the training data contains some examples of compositions, models can indeed generalise to completely new compositions.

| Grounded Concept | Size | Example words |
|---|---|---|
| Cardinal directions | 4 | north, south, east, west |
| + compositions | 4 | northeast, southeast, northwest, southwest |
| Spatial directions | 8 | left, right, up, down, top, bottom, above, below |
| Colours | 107 | red, blue, yellow, green, violet, aqua, wine, charcoal, brass, cobalt.. |
| + *antique* modifier | 4 | antique brass, antique fuchsia, antique ruby, antique white |
| + *baby* modifier | 3 | baby blue, baby blue eyes, baby pink |
| + *bright* modifier | 7 | bright cerulean, bright green, bright lavender, bright maroon |
| + *burnt* modifier | 3 | burnt orange, burnt sienna, burnt umber |
| + *copper* modifier | 5 | copper crayola, copper penny, copper red, copper rose, pale copper |
| + *dark* modifier | 43 | dark blue, dark brown, dark byzantium, dark cerulean, dark chestnut.. |
| + *deep* modifier | 17 | deep carmine, deep carmine pink, deep carrot orange, deep cerise.. |
| + *electric* modifier | 14 | electric blue, electric crimson, electric cyan, electric indigo.. |
| + *fluorescent* modifier | 3 | fluorescent orange, fluorescent pink, fluorescent yellow |
| + *french* modifier | 6 | french beige, french blue, french lilac, french lime, french raspberry.. |
| + *light* modifier | 24 | light apricot, light blue, light brown, light carmine pink, light coral.. |
| + *medium* modifier | 21 | medium aquamarine, medium blue, medium carmine.. |
| + *old* modifier | 5 | old gold, old lace, old lavender, old mauve, old rose |
| + *pale* modifier | 20 | pale aqua, pale blue, pale brown, pale carmine, pale cerulean.. |
| + *pastel* modifier | 16 | dark pastel blue, dark pastel green, dark pastel purple, dark pastel red.. |
| + *persian* modifier | 9 | persian blue, persian green, persian indigo, persian orange |
| + *rich* modifier | 7 | rich black, rich brilliant lavender, rich carmine, rich lavender.. |
| + *rose* modifier | 12 | copper rose, french rose, old rose, persian rose, rose bonbon |
| + *royal* modifier | 6 | royal azure, royal blue web, royal fuchsia, royal purple, royal yellow |
| + *vivid* modifier | 5 | vivid auburn, vivid burgundy, vivid cerise, vivid tangerine, vivid violet |

Table 7: Table shows the grounded concepts we aim to teach language models. The first column shows the category of related concepts that models might have learnt relational meaning for, the second shows the number of such terms within each category, and the third shows example words.

## C   EVALUATION METRICS

In this section we describe our evaluation metrics in detail, with examples of how certain generations might be scored by each metric.

### C.1   SUBSTRING MATCH

For every test instance, there is exactly one string answer that is correct (for e.g., the words *"left"* or *"northeast"* or *"electric blue"*). However, language models are free to generate any number of tokens given a prompt—i.e., they might generate exactly one token, or up to a 100 tokens for any given prompt. We consider the first 5 tokens generated by the model to be the generated answer— and consider a substring match metric to be one that looks at whether or not the model generated answer lies in the ground truth answer. To make this clearer, we provide examples below, denoting a model-generated answer in blue and the ground-truth answer in green. By the substring metric, electric green for electric blue would give an accuracy of 0, however electric or green would give an accuracy of 1. In practice, we do not see (the large) models generate half-answers (e.g,. simply saying *electric*). Similarly green for dark green has an accuracy of 1, but dark pink for dark green would have an accuracy of 0.

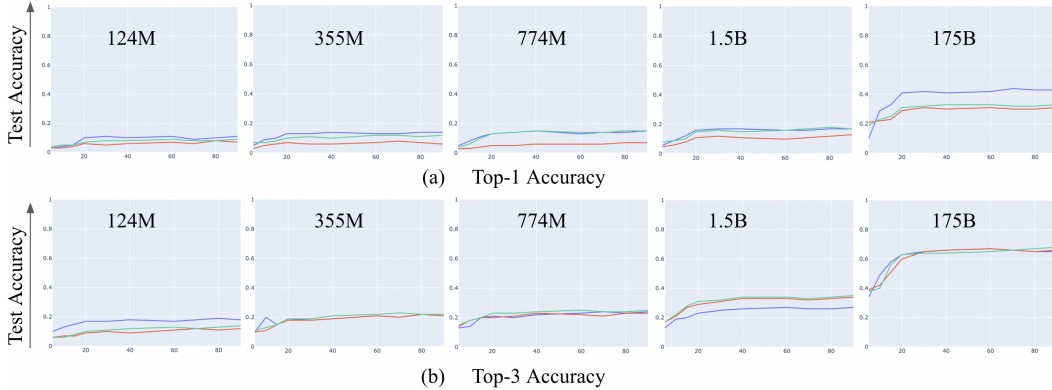

Figure 5: Figure shows performance curves of models on increasing the number of samples in a prompt. From left to right, we see the 124M, 355M, 774M, 1.5B and 175B parameter models, on increasing the number of samples given to the model to the maximum prompt size. We see that only 5 samples are not enough for models to learn the task, but past a certain threshold (20 samples), models do not have a significant increase in performance on the same test set.

## C.2 EXACT MATCH

For this metric, the model only gets a perfect accuracy if the model generated answer exactly matches the ground-truth answer (after stripping both for any trailing whitespaces). Therefore saying green for dark green , or electric green for light electric green would have an accuracy of 0.

## C.3 GROUNDING DISTANCE

Equation C.3 shows our calculation of distances in the world. For a certain concept category $C$ (for example, all colour names existing in the dataset), let $c_1$ be the model predicted concept name and $c_2$ be the true grounding. The grounding distance is the Euclidean distance between the two points when the predicted concept does exist in the space of concepts in the world, and is set to an arbitrarily high number (here, 500, which is higher than the maximum distance between any two points in space) when the predicted concept is some other word does not fall in the concept category.

$$
\mathrm{d}(c_1, c_2) = \begin{cases} \sqrt{(c_{1x} - c_{2x})^2 + (c_{1y} - c_{2y})^2 + (c_{1z} - c_{2z})^2} & \text{for} \quad c_1 \in C \\ 500 & \text{for} \quad c_1 \notin C \end{cases}
$$

## D PROMPT SIZE

The size of the input prompt to a model depends on its maximum content length, which is a hyper-parameter fixed before training the model to completion. The question-answer pairs that we use to teach models grounded concepts differ in their lengths based on the concept category. For example, since the spatial and cardinal concepts require grid worlds that could be up to 10 rows and columns wide, a prompt for these categories might contain a fewer number of samples. Since the colour groundings are 3-digit RGB codes, a larger number of samples can be given in a prompt. We report accuracy curves of models when given an increasing number of samples in the prompt in Appendix D. We see that past 20 samples for spatial terms, and 60 samples for colours, models have no significant increase in performance. When we report results in Tables ?? and Table 1, we report numbers for a fixed number of samples in a prompt (e.g., 20 vs. 60 respectively) for all models.

Interestingly, once we get past a certain number of prompts, there is no significant increase in model performance on the task. This result hints at two things. First, the larger models seem to learn

a task, and generalise to new samples, with only a small number of sample when fine-tuned in a few-shot prompting regime. This seems to hold for the smaller models as well i.e., although they do not learn the task well, increasing the number of samples within a prompt does not significantly increase model performance. Second, the significant difference in performance based on model size hints at the fact that the limiting factor for good performance is the number of parameters of a model. More concretely, the generalisation extent of each model seems to max out at some number of input prompts, with a trend of increasing performance on increasing model size. Therefore, a larger model (than the ones we have here) might be required in order to achieve better grounded generalisation.

## E  MODEL GENERATIONS

We show example generations from the model in Table 8.

| Category | Prompt | True | 124M | 355M | 774M | 1.5B | 175B |
|---|---|---|---|---|---|---|---|
| Cardinal | **World:** [0. 0. 0. 0.]
[0. 0. 0. 0.]
[0. 0. 0. 1.] | southeast | world | direction | the | south | southeast |
| | **World:** [0. 0. 1. 0.]
[0. 0. 0. 0.] | north | the | direction | world | south | north |
| Spatial | **World:** [1. 0. 0. 0.]
[0. 0. 0. 0.]
[0. 0. 0. 0.] | left | the | world | an | left | left |
| | **World:** [0. 1.]
[0. 0.]
[0. 0.]
[0. 0.] | right | and | world | to | world | right |
| Colour | **RGB:** (123, 0, 43) | light blue | color | the | rgb | red | blue |
| | **RGB:** (3, 43, 100) | magenta | hex | red | color | green | red |

Table 8: Table shows example generations from each of the models, cut off for the first five words. The first column shows the concept category while the second column shows the last portion of the prompt given to each model, for which they are required to generate an answer. The last 6 columns show the ground-truth answer, and predicted answers from each of the models respectively.

## F  PRE-TRAINING DATA

### F.1  WHAT HAVE MODELS SEEN IN THEIR TRAINING DATA THAT MIGHT BE RELATED TO GROUNDING?

Pre-trained language models have been trained on large text corpora available on the internet—a diverse source of information that covers many domains. There is speculation, that the remarkable generalisation performance of these models stems from having seen instances of the task or related information at some point in their training data. However, this data has not been made public, which makes this a hypothesis that is hard to confirm. In our domain of grounded concepts as well, it is unclear what aspects of similar data the model might have had access to during training, or what inductive biases from certain types of data it might have learnt, that allow it good performance on grounded tasks. We attempt to assess this in the following way. We consider all the concept words in every concept category, and all the world representations, and consider these the prompts for the models. For each word, we wish to assess the distribution of text generated, which should, in some sense, reflect the distribution of text seen during training. For example, or the world representations (i.e., a 2D array), we assess how often the model might use a spatial or cardinal term when simply prompted with an array, thus giving us insights into how often the training data might have contain such an array in close proximity to such a word. Similarly, for the colour domain, we evaluate how often the model predicts a correct colour name when prompted with an RGB code. We see that when models are simply prompted with a world representation, they tend to generate similar-

looking world representations, instead of conceptual words (such as *"left"*) that might be associated with it. When prompted with RGB codes, the GPT-3 model only has a 7.3% accuracy of correctly generating the corresponding colour name in a 5-token sequence after the prompt.

|  |  | **Cardinal** | | |  |  | **Cardinal** | | |
|---|---|---|---|---|---|---|---|---|---|
|  |  | Original | Rotated | Random |  |  | Original | Rotated | Random |
|  | **R** | 0.00 | 0.00 | 0.00 |  | **R** | 0.00 | 0.00 | 0.00 |
|  | **R\*** | **0.13** | **0.13** | **0.13** |  | **R\*** | 0.13 | 0.13 | 0.13 |
| **Top-1 Accuracy** | **124 M** | 0.03 | 0.02 | 0.02 | **124 M** | | 0.11 | 0.10 | 0.05 |
|  | **355 M** | 0.03 | 0.04 | 0.03 | **355 M** | | 0.10 | 0.11 | 0.06 |
|  | **774 M** | 0.04 | 0.03 | 0.02 | **774 M** | | 0.13 | 0.12 | 0.08 |
|  | **1.5 B** | 0.07 | 0.06 | 0.05 | **1.5 B** | | 0.13 | 0.14 | 0.10 |
|  | **175 B** | 0.12 | 0.11 | 0.10 | **175 B** | | **0.30** | **0.29** | **0.08** |
|  | **R-IV** | 0.00 | 0.00 | 0.00 |  | **R-IV** | 0.00 | 0.00 | 0.00 |
|  | **R-ID** | 0.13 | 0.13 | 0.13 |  | **R-ID** | 0.13 | 0.13 | 0.13 |
| **Top-3 Accuracy** | **124 M** | 0.03 | 0.04 | 0.04 | **124 M** | | 0.09 | 0.09 | 0.08 |
|  | **355 M** | 0.06 | 0.05 | 0.04 | **355 M** | | 0.19 | 0.17 | 0.10 |
|  | **774 M** | 0.05 | 0.07 | 0.05 | **774 M** | | 0.17 | 0.15 | 0.11 |
|  | **1.5 B** | 0.11 | 0.10 | 0.07 | **1.5 B** | | 0.21 | 0.20 | 0.14 |
|  | **175 B** | **0.23** | **0.21** | **0.17** | **175 B** | | **0.60** | **0.61** | **0.09** |

Table 9: Table shows evaluations in the cardinal directions domain, where we show models examples of single directions (*north, south, east, west*) and test them on compositions of directions (*northeast, southeast, northwest, southwest*) that were never seen before. Rows show model sizes for GPT-2 models (124M to 1.5B) and the 175B GPT-3 model. Columns show metrics for the original, rotated and random worlds for each of the concept categories. We see that all models fail on this task i.e., when they have never seen compositions of terms before, they never predict them at test-time.

Table 10: In this table, we show models examples of concepts within a sub-space (*north, east, northeast*) and test them on concepts in a different sub-space (*south, west, southwest*) that were never seen before. Rows show model sizes for GPT-2 models (124M to 1.5B) and the 175B GPT-3 model. Columns show metrics for the original, rotated and random worlds for each of the concept categories. We see that the largest model can indeed perform on this task i.e., even on only having seen a sub-space of the world (along with compositions) it can generalise to a new sub-space and compositions.

# G    EXPERIMENTS WITH MASKED LANGUAGE MODELS

The results in the main paper focus on GPT-2 and GPT-3 language models i.e., autoregressive left-to-right language models that show impressive capabilities for in-context learning. Here, we provide comparisons to a masked language model, specifically a BERT model (Devlin et al., 2018). For our implementation we use a BERT-base model containing 110 million parameters. Similar to the GPT-x experiments for in-context learning with prompts, we "teach" the BERT model the space on concepts in the following way.

- **BERT zero-shot (no-context)** In this setting, for every sample in the test set (e.g., an RGB colour), we create an input prompt that masks the name of the colour, and ask the model to fill in the colour name. The prompt, therefore, would look like `RGB: (255, 0, 0) Colour: <mask>`. For the same test set, we evaluate the Top-1 and Top-3 accuracy of the model filling in the mask with the correct colour, using the same substring-match metric.
- **BERT zero-shot (in-context)** In this setting, we add several samples into the context of the model, and ask the model to fill in the masked token with the colour name, for all samples in the test set. Since the context-width of BERT is smaller than the GPT models, we can only include a smaller number of samples (around 7).

| | | Spatial | | | Cardinal | | | Colours | | |
|---|---|---|---|---|---|---|---|---|---|---|
| | | Original | Rotated | Random | Original | Rotated | Random | Original | Rotated | Random |
| | **R-IV** | 0.00 | 0.00 | 0.00 | 0.00 | 0.00 | 0.00 | 0.00 | 0.00 | 0.00 |
| | **R-ID** | 0.16 | 0.16 | **0.16** | 0.13 | 0.13 | 0.13 | 0.00 | 0.00 | 0.00 |
| | **124 M** | 0.10 | 0.11 | 0.04 | 0.10 | 0.10 | 0.03 | 0.09 | 0.09 | 0.10 |
| **Top-1** | **355 M** | 0.10 | 0.11 | 0.06 | 0.08 | 0.07 | 0.03 | 0.13 | 0.12 | 0.10 |
| **Accuracy** | **774 M** | 0.09 | 0.11 | 0.03 | 0.10 | 0.09 | 0.01 | 0.11 | 0.12 | 0.09 |
| | **1.5 B** | 0.14 | 0.14 | 0.12 | 0.13 | 0.12 | 0.11 | 0.16 | 0.15 | 0.11 |
| | **175 B** | **0.28** | **0.27** | 0.13 | **0.29** | **0.28** | **0.15** | **0.42** | **0.41** | **0.14** |
| | **R-IV** | 0.00 | 0.00 | 0.00 | 0.00 | 0.00 | 0.00 | 0.00 | 0.00 | 0.00 |
| | **R-ID** | 0.16 | 0.16 | 0.16 | 0.13 | 0.13 | 0.13 | 0.00 | 0.00 | 0.00 |
| | **124 M** | 0.13 | 0.12 | 0.07 | 0.10 | 0.10 | 0.08 | 0.16 | 0.14 | 0.10 |
| **Top-3** | **355 M** | 0.24 | 0.17 | 0.15 | 0.20 | 0.18 | 0.13 | 0.25 | 0.24 | 0.10 |
| **Accuracy** | **774 M** | 0.19 | 0.24 | 0.12 | 0.21 | 0.21 | 0.17 | 0.23 | 0.19 | 0.11 |
| | **1.5 B** | 0.32 | 0.29 | 0.20 | 0.29 | 0.27 | 0.20 | 0.29 | 0.32 | 0.23 |
| | **175 B** | **0.64** | **0.65** | **0.21** | **0.63** | **0.64** | **0.16** | **0.63** | **0.62** | **0.12** |

Table 11: Table shows evaluations for generalisation to unseen concepts for a random split of training/test data. Rows show model sizes for GPT-2 models (124M to 1.5B) and the 175B GPT-3 model. Columns show metrics for the original, rotated and random worlds for each of the concept categories. We report both Top-1 and Top-3 accuracy. R-IV refers to a random baseline that randomly selects a word out of the total vocabulary, R-ID refers to a baseline that randomly selects a word out of the set of words in the concept category.

| | | Spatial | | | Cardinal | | | Colours | | |
|---|---|---|---|---|---|---|---|---|---|---|
| | | Original | Rotated | Random | Original | Rotated | Random | Original | Rotated | Random |
| | **R-IV** | 0.00 | 0.00 | 0.00 | 0.00 | 0.00 | 0.00 | 0.00 | 0.00 | 0.00 |
| | **R-ID** | 0.16 | 0.16 | **0.16** | 0.13 | 0.13 | **0.13** | 0.00 | 0.00 | 0.00 |
| | **124 M** | 0.00 | 0.00 | 0.00 | 0.01 | 0.00 | 0.00 | 1.00 | 0.00 | 1.00 |
| **Top-1** | **355 M** | 0.00 | 0.01 | 0.00 | 0.02 | 0.02 | 0.01 | 0.02 | 0.03 | 0.01 |
| **Accuracy** | **774 M** | 0.00 | 0.00 | 0.01 | 0.00 | 0.01 | 0.00 | 0.02 | 0.01 | 0.01 |
| | **1.5 B** | 0.01 | 0.00 | 0.01 | 0.01 | 0.03 | 0.00 | 0.04 | 0.02 | 0.00 |
| | **175 B** | 0.12 | 0.14 | 0.07 | **0.14** | **0.15** | 0.06 | **0.19** | **0.18** | **0.08** |
| | **R-IV** | 0.00 | 0.00 | 0.00 | 0.00 | 0.00 | 0.00 | 0.00 | 0.00 | 0.00 |
| | **R-ID** | 0.16 | 0.16 | **0.16** | 0.13 | 0.13 | **0.13** | 0.00 | 0.00 | 0.00 |
| | **124 M** | 0.01 | 0.00 | 0.01 | 0.00 | 0.02 | 0.00 | 0.03 | 0.03 | 0.01 |
| **Top-3** | **355 M** | 0.04 | 0.03 | 0.00 | 0.04 | 0.04 | 0.01 | 0.07 | 0.05 | 0.03 |
| **Accuracy** | **774 M** | 0.05 | 0.04 | 0.04 | 0.03 | 0.04 | 0.00 | 0.05 | 0.05 | 0.02 |
| | **1.5 B** | 0.09 | 0.09 | 0.07 | 0.07 | 0.05 | 0.01 | 0.10 | 0.09 | 0.04 |
| | **175 B** | **0.24** | **0.25** | 0.09 | **0.21** | **0.23** | **0.13** | **0.36** | **0.36** | **0.11** |

Table 12: Table shows evaluations for generalisation to unseen concepts using an exact substring-match evaluation criteria. Rows show model sizes for GPT-2 models (124M to 1.5B) and the 175B GPT-3 model. Columns show metrics for the original, rotated and random worlds for each of the concept categories. We report both Top-1 and Top-3 accuracy. R-IV refers to a random baseline that randomly selects a word out of the total vocabulary, R-ID refers to a baseline that randomly selects a word out of the set of words in the concept category. This table is equivalent to Table F.1, except using an exact-match metric instead of substring-match.
This table is equivalent to Table 1, except using an exact-match metric instead of substring-match.

- **BERT zero-shot (fine-tuned)** In this setting, we fine-tune the BERT model on 67 samples (the same number of samples that the GPT models got as input in a prompt) and then report performance when tested on all the colours in the test set.

Table 13 shows results of the BERT models when tested on a held-out sub-space of colours and Table 14 shows results when tested on a randomly held-out split of colours.

| | | Colour | | |
|---|---|---|---|---|
| | | Original | Rotated | Random |
| **Top-1 Accuracy** | **R** | 0.00 | 0.00 | 0.00 |
| | **R*** | 0.13 | 0.13 | 0.13 |
| | **124 M** | 0.08 | 0.09 | 0.03 |
| | **355 M** | 0.06 | 0.07 | 0.04 |
| | **774 M** | 0.11 | 0.09 | 0.01 |
| | **1.5 B** | 0.10 | 0.09 | 0.06 |
| | **175 B** | 0.23 | 0.21 | 0.11 |
| | **BERT** zero-shot (no-context) | 0.09 | 0.08 | 0.08 |
| | **BERT** zero-shot (in-context) | 0.08 | 0.07 | 0.05 |
| | **BERT** fine-tuned | 0.10 | 0.09 | 0.07 |
| **Top-3 Accuracy** | **R-IV** | 0.00 | 0.00 | 0.00 |
| | **R-ID** | 0.13 | 0.13 | 0.13 |
| | **124 M** | 0.06 | 0.05 | 0.04 |
| | **355 M** | 0.14 | 0.11 | 0.12 |
| | **774 M** | 0.15 | 0.16 | 0.14 |
| | **1.5 B** | 0.19 | 0.18 | 0.16 |
| | **175 B** | 0.34 | 0.36 | 0.13 |
| | **BERT** zero-shot (no-context) | 0.10 | 0.09 | 0.06 |
| | **BERT** zero-shot (in-context) | 0.11 | 0.09 | 0.07 |
| | **BERT** fine-tuned | 0.11 | 0.10 | 0.09 |

Table 13: Table shows evaluations for the BERT model when tested in 3 different ways. in the colour domain, where we show models examples of colours in a sub-space (e.g., all colours in a sphere of Euclidean distance 150 from *red*) and test them on different subspaces i.e., all remaining colours. Rows show model sizes for GPT-2 models (124M to 1.5B) and the 175B GPT-3 model. Columns show metrics for the original, rotated and random worlds for each of the concept categories.

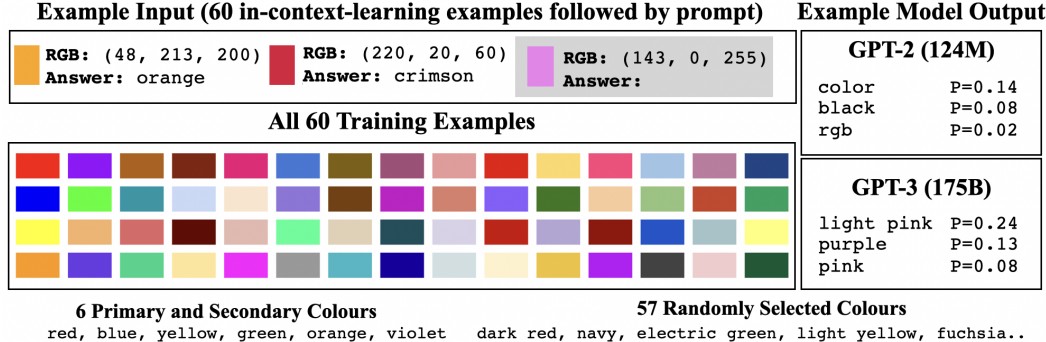

Figure 6: Figure shows example worlds and groundings for the colour domain. The left panel shows example RGB codes and associated colour names, while the right shows example outputs from the smallest and largest models, with the top three most probable words from each model along with the probability of that word. When computing Top-1 vs. Top-3 accuracy, we consider only the first vs. all 3 outputs respectively.

|  |  | Colour | | |
|---|---|---|---|---|
|  |  | Original | Rotated | Random |
|  | **R** | 0.00 | 0.00 | 0.00 |
|  | **R\*** | 0.00 | 0.00 | 0.00 |
| **Top-1 Accuracy** | **124 M** | 0.09 | 0.09 | 0.10 |
|  | **355 M** | 0.13 | 0.12 | 0.10 |
|  | **774 M** | 0.11 | 0.12 | 0.09 |
|  | **1.5 B** | 0.16 | 0.15 | 0.11 |
|  | **175 B** | 0.42 | 0.41 | 0.14 |
|  | **BERT** zero-shot (no-context) | 0.03 | 0.05 | 0.04 |
|  | **BERT** zero-shot (in-context) | 0.04 | 0.05 | 0.05 |
|  | **BERT** fine-tuned | 0.08 | 0.09 | 0.10 |
|  | **R-IV** | 0.00 | 0.00 | 0.00 |
|  | **R-ID** | 0.00 | 0.00 | 0.00 |
| **Top-3 Accuracy** | **124 M** | 0.16 | 0.14 | 0.10 |
|  | **355 M** | 0.25 | 0.24 | 0.10 |
|  | **774 M** | 0.23 | 0.19 | 0.11 |
|  | **1.5 B** | 0.29 | 0.32 | 0.23 |
|  | **175 B** | 0.63 | 0.63 | 0.12 |
|  | **BERT** zero-shot (no-context) | 0.10 | 0.09 | 0.09 |
|  | **BERT** zero-shot (in-context) | 0.09 | 0.09 | 0.07 |
|  | **BERT** fine-tuned | 0.10 | 0.11 | 0.09 |

Table 14: Table shows evaluations for the BERT model when tested in 3 different ways. in the colour domain, where we show models examples of colours from a random split of 70 colours and test them on all remaining colours. Rows show model sizes for GPT-2 models (124M to 1.5B), the 175B GPT-3 model, and the BERT-base model when given samples in 3 different ways. Columns show metrics for the original, rotated and random worlds for each of the concept categories.

|  |  | Spatial | | | Cardinal | | | Colours | | |
|---|---|---|---|---|---|---|---|---|---|---|
|  |  | Original | Rotated | Random | Original | Rotated | Random | Original | Rotated | Random |
|  | **R-IV** | 0.00 | 0.00 | 0.00 | 0.00 | 0.00 | 0.00 | 0.00 | 0.00 | 0.00 |
|  | **R-ID** | 0.16 | 0.16 | **0.16** | 0.13 | 0.13 | **0.13** | 0.00 | 0.00 | 0.00 |
| **Top-1 Accuracy** | **124 M** | 0.00 | 0.01 | 0.00 | 0.01 | 0.00 | 0.00 | 0.02 | 0.01 | 1.00 |
|  | **355 M** | 0.02 | 0.01 | 0.00 | 0.03 | 0.02 | 0.01 | 0.03 | 0.03 | 0.01 |
|  | **774 M** | 0.02 | 0.02 | 0.01 | 0.03 | 0.02 | 0.00 | 0.07 | 0.06 | 0.03 |
|  | **1.5 B** | 0.06 | 0.07 | 0.03 | 0.07 | 0.07 | 0.05 | 0.09 | 0.09 | 0.05 |
|  | **175 B** | 0.09 | 0.09 | 0.04 | 0.10 | 0.09 | 0.05 | **0.19** | **0.20** | **0.08** |
|  | **R-IV** | 0.00 | 0.00 | 0.00 | 0.00 | 0.00 | 0.00 | 0.00 | 0.00 | 0.00 |
|  | **R-ID** | 0.16 | 0.16 | **0.16** | 0.13 | 0.13 | **0.13** | 0.00 | 0.00 | 0.00 |
| **Top-3 Accuracy** | **124 M** | 0.01 | 0.01 | 0.00 | 0.02 | 0.03 | 0.01 | 0.05 | 0.05 | 0.02 |
|  | **355 M** | 0.04 | 0.04 | 0.02 | 0.03 | 0.04 | 0.02 | 0.06 | 0.05 | 0.04 |
|  | **774 M** | 0.04 | 0.05 | 0.04 | 0.03 | 0.04 | 0.00 | 0.10 | 0.09 | 0.03 |
|  | **1.5 B** | 0.09 | 0.09 | 0.04 | 0.09 | 0.10 | 0.05 | 0.12 | 0.11 | 0.06 |
|  | **175 B** | **0.30** | **0.31** | 0.10 | **0.35** | **0.32** | **0.13** | **0.26** | **0.25** | **0.10** |

Table 15: Table shows evaluations for generalisation to unseen concepts using an exact substring-match evaluation criteria. Rows show model sizes for GPT-2 models (124M to 1.5B) and the 175B GPT-3 model. Columns show metrics for the original, rotated and random worlds for each of the concept categories. We report both Top-1 and Top-3 accuracy. R-IV refers to a random baseline that randomly selects a word out of the total vocabulary, R-ID refers to a baseline that randomly selects a word out of the set of words in the concept category. This table is equivalent to Table 2, except using an exact-match metric instead of substring-match.

| | | Top-1 Accuracy | | | | | | Top-3 Accuracy | | | | |
| --- | --- | --- | --- | --- | --- | --- | --- | --- | --- | --- | --- | --- |
| | **Spatial** | | | **Cardinal** | | | **Spatial** | | | **Cardinal** | | |
| | Orig. | Rot. | Rand. | Orig. | Rot. | Rand. | Orig. | Rot. | Rand. | Orig. | Rot. | Rand. |
| **R-IV** | 0.00 | 0.00 | 0.00 | 0.00 | 0.00 | 0.00 | 0.00 | 0.00 | 0.00 | 0.00 | 0.00 | 0.00 |
| **R-ID** | 0.16 | 0.16 | 0.16 | 0.13 | 0.13 | 0.13 | 0.16 | 0.16 | 0.16 | 0.13 | 0.13 | 0.13 |
| **124 M** | 0.02 | 0.02 | 0.02 | 0.03 | 0.03 | 0.02 | 0.03 | 0.02 | 0.01 | 0.02 | 0.02 | 0.01 |
| **355 M** | 0.03 | 0.02 | 0.03 | 0.03 | 0.03 | 0.02 | 0.04 | 0.05 | 0.04 | 0.04 | 0.03 | 0.03 |
| **774 M** | 0.07 | 0.05 | 0.06 | 0.06 | 0.07 | 0.04 | 0.09 | 0.07 | 0.05 | 0.08 | 0.08 | 0.04 |
| **1.5 B** | 0.10 | 0.11 | 0.11 | 0.07 | 0.08 | 0.06 | 0.08 | 0.07 | 0.07 | 0.09 | 0.08 | 0.08 |
| **175 B** | **0.39** | **0.39** | **0.13** | **0.38** | **0.39** | **0.14** | **0.49** | **0.50** | **0.15** | **0.58** | **0.59** | **0.18** |

Table 16: Table shows evaluations for generalisation to unseen worlds. Rows show model sizes for GPT-2 models (124M to 1.5B) and the 175B GPT-3 model. Columns show metrics for the original, rotated and random worlds for each of the concept categories. We report both Top-1 and Top-3 accuracy in the first and second columns of each rotated world. This table is equivalent to Table 1, except using an exact-match metric instead of substring-match.

