# OpenReview forum: "Mapping Language Models to Grounded Conceptual Spaces"
_ICLR.cc/2022/Conference — ICLR 2022 Poster_

### Official Review · Reviewer_qf1D · 2021-11-02

**Correctness:** 3
**Technical Novelty And Significance:** 3
**Empirical Novelty And Significance:** 4
**Recommendation:** 5
**Confidence:** 5

**Main Review:**

## Strengths

* I think this is a nice set of experiments. The choices to use both unseen worlds and unseen concepts was good. The rotated baseline is also a great idea, and frankly I find the results on rotated spaces quite surprising, especially when it comes to color. I would have expected a serious performance decrease. It speaks very well to the model's ability to construct an alignment between the spaces.

* I think the framing and conceptual situating this paper does is pretty good. The discussion of the difficulty of "grounding" complex concepts that can't be encoded easily in text is important and I'm glad it was included. The principal theoretical complaint that I foresee people having regarding the overall framing of the paper is that it's not "really" testing grounding because of the way these the concepts are serialized into text, but I think the authors are very clear about how this approach relies on the faithful encoding of a space into text and that it's a serious limitation particularly when it comes to more complex concepts.

## Weaknesses

I think there are some subtle but pretty important problems with the paper. To fix them, it might be okay to just change the framing and back off from the "isomorphism" angle, but that feels like losing the whole point of the paper and leaving it kind of uninteresting. On the other hand, I think maybe adding some tougher experiments and better baselines could patch up the holes and make the paper really nice. Details follow:

* Abdou et al. is mentioned as a data source, but I think deeper comparisons and more credit to that work should be provided. This paper presents itself as providing clear evidence of learning "isomorphic" conceptual spaces from text alone, I think the Abdou et al. paper makes a case that is very similar (though meaningfully different, yes) and stronger than this one. At the very least, these results should be put in context of theirs (especially as their data is being used). In particular the Abdou et al. paper already provides pretty good evidence of conceptual grounding of colors using representational similarity analysis and linear mappings on color representations extracted from the LMs. These methods have some underlying assumptions but it seems to me that those assumptions are much weaker than the ones underlying the method used in this paper, which relies on the few-shot learning capabilities of the language model. More on that in the next point.

* I don't think the R-IV baseline is more fair than the R-ID one, as claimed in Sec. 2.5. This paper is about producing the *correct* term for the provided grounding, which is a different problem from producing an *in-domain* term. Generating an in-domain term is *a priori* significantly easier (e.g., by hard-coding it). By including the requirement to generate an in-domain term, these experiments conflate the two problems. So GPT-2, for example, does badly because in the few-shot format, it often can't decide on any answer with high enough confidence to make it to the top of the beam (see in Figure 3 how GPT-2 skips giving an answer at all and just moves on to the next prompt). This is a classic example of "boring" sequence generation that happens for sequence models over high entropy output distributions (the literature on this focuses generally focuses on beam search, but I think the argument can reasonably apply in cases like this; see [Holtzman et al. 2019](https://openreview.net/forum?id=rygGQyrFvH), [Stahlberg and Byrne 2019](https://aclanthology.org/D19-1331/), [Eikema and Aziz 2020](https://aclanthology.org/2020.coling-main.398.pdf)), which is an orthogonal issue to the grounding problem. The issue here is a combination of that and the mere fact that the smaller GPT-2 models don't make sense of few-shot prompts in the effective way that GPT-3 does — again, arguably a separate issue from grounding. In short, by relying on the few-shot learning paradigm to extract the conceptual structure of concern, we're limited to extracting conceptual structure that can surface _through few-shot learning_ (just like, for example, linear probes can only extract linearly separable features). This ends up being more a measure of the few-shot learning ability of the model than the conceptual grounding capabilities of the model. This is particularly problematic because it makes the baselines (ie, GPT-2 models) seem weaker than they probably should be, and it limits the baselines that can be used. I would point particularly to Abdou et al., who obtained alignments similar in quality to those from masked LMs (BERT, RoBERTa, ELECTRA) using just PMI statistics and fastText embeddings. It's not clear to me if the large models are exhibiting conceptual generalization/grounding abilities much larger than those afforded by such baselines (once we move out of the few-shot learning format).

* To add to this issue, the headline experiment of generalizing to new concepts has at least one big issue that I find concerning. Including *all* n-1 concepts besides the target one, or many such concepts (eg, 60 colors) provides a very strong constraint in the input (in my view contradicting the "only a small number of examples" claim in the paper's abstract). While, yes, generalizing to new concepts does demonstrate that the model understands something about the domain, I think it's important that we understand this in graded terms. The question this paper seems designed to address, based on the framing in the introduction and conclusion, is about whether language models can or do construct conceptual spaces that are roughly "isomorphic" to grounded ones. In order to test this, of course, we have to construct the isomorphism. But — and this is one of the classic problems with probing — if we are looking at it in terms of set isomorphism (i.e., without extra homomorphism structure — and we are, if we're measuring prediction accuracy), then it is always possible to reconstruct such an isomorphism if we have enough supervision (See [Pimentel et al.](https://arxiv.org/abs/2004.03061)). The question is *how much* supervision we need versus how much is usefully obtainable from the pretrained model, hence approaches like [Voita and Titov (2020)'s](https://aclanthology.org/2020.emnlp-main.14.pdf) information-theoretic probing. So it's not really a black and white question whether there is an "isomorphic" conceptual space inside the model, but a graded one. And when we talk about things in terms of an isomorphism, what we're really implying is that the vast majority of the information required to construct the isomorphism is easily accessible from the model representations, so less supervision is required. This is the basic thinking behind the idea of a "minimal grounding toehold" which has been discussed as a prerequisite to grounding language models in conceptual spaces. Some of this discussion in the NLP community subsequent to Bender and Koller's 2020 paper was catalogued in [a blog post](https://blog.julianmichael.org/2020/07/23/to-dissect-an-octopus.html) that goes into a lot more detail on the issue. That post proposed a color experiment extremely close to what is done in this paper, but an important difference in their proposal is that the input used by the LM to reconstruct the grounded lexicon is the *minimal* size that is required to span the conceptual space (i.e., constraining only the *necessary* degrees of freedom). One might use light/dark, red/green, and blue/yellow as the three basis dimensions to align with CIELAB, or even just to try red, green, and blue. I think such a minimal experiment would be required to effectively address the isomorphism question. This paper, by providing many more inputs, is doing something of a mix of probing (because of the few-shot/supervision aspect) and behavioral testing (because it's relying on inherent generalization behavior in the LM distribution).
  * [Edit Nov 3: One extra thought.] The inclusion of a bunch of distinct concepts in the input can potentially also provide the model with a pragmatic cue that it will be presented with a new/unseen concept label if it receives a new/unseen kind of input, which adds another confound. This may not be much of an issue in the case of color since the space is fairly open and only 1/6 of the set of colors was covered anyway, but it still means a totally ungrounded baseline could do better than random.

To sum up: to really address the question of interest in this paper, I think the unseen-concept experiments need to more strongly limit their inputs to a very small number (or even better, report curves showing performance dependent on number of inputs), LM experiments which constrain the output space should be included as well, and predictive baselines outside of the few-shot paradigm should be provided. The results of Abdou et al. seem to indicate (to me) that it should be possible to do quite a bit better than the GPT-2 results in this paper using some kind of simple model (for example with word embeddings or a probing model on LM representations). It's important to have these baselines so as not to misrepresent the performance trend with respect to model growth, which I suspect reflects more the ability of the model to do "few-shot learning" than to ground concepts.

(And to be clear: I don't think it would be a strike against this paper if few-shot learning for unseen concepts with very few colors didn't give good performance. That's still a cool experiment and an interesting result!)

Relatively minor issues:

* I understand you would probably want to express the color in RGB for the purposes of LM testing, but
Rotations in RGB seems like it isn't the best way of doing the isomorphism control: rotating in a space where distances are more perceptually meaningful (such as CIELAB) seems to me like it would have been a better approach, as while rotation preserves distances, the same distances in different parts of RGB space may correspond to different perceptual distances.

* On that note, I'm not sure if I totally understand the random world baseline. It sounds from the description in the text (Sec. 2.4) that you ground words entirely randomly, but Figure 2 illustrates a "random rotation" which is much more constrained than this. It suggests that a random rotation should destroy the underlying structure, but I don't see why this would be the case. As long as distances are preserved (under the assumption necessary for the control that distances are meaningful in the first place) the structure should be preserved. So any rotation should work. In what sense does a random rotation destroy the structure? I don't buy it based just on the example shown in Figure 2.

* The string matching Top-1 accuracy metric seems quite permissive — are there examples, or is there a human verification step you can provide to show that it is reasonable in this case and not overly permissive?

* There seems to be some confusion about the training data. When you first mention it, you use the term OPENAI-WT with no further description or reference (Sec. 2.1). I don't think that is the dataset's name — I found it used as an abbreviation in one paper, Gehman et al. 2020, but nowhere else. The dataset is just called WebText. Can you just call it WebText (or OpenAI WebText) and cite the GPT-2 paper directly when you name it (so it's clear where it's from)? Also, in Section 2.1 it says all models are trained on this data, but this is incorrect — GPT-3 was trained on a different mix of data including two corpora of books, Wikipedia, and Common Crawl. Also, Appendix A calls the data "OpenWT", which sounds like "OpenWebText", which is an open dataset scraped using the same methods as WebText — but isn't the same dataset as was used to train GPT-2. Then, Section 2.4 does cite the GPT-2 paper for the data, but calls it CommonCrawl, which is incorrect — that's a different web scrape.

## Typos etc.

* Sec. 2.2, fourth line: stray space after "prompt"
* End of page 2: stray comma after "detail"
* The citation to Abdou et al. can be updated — the work is appearing in CoNLL 2021.
* Last sentence before Sec. 3: stray comma at "R-IV, a fairer"

**Summary Of The Paper:**

This paper investigates the question of whether language models encode, in some way or another, conceptual spaces that are analogous or isomorphic to grounded ones. The paper reports on the design and results of several experiments to test this in domains which can be easily serialized as text (and hence made accessible to an LM) — spatial directional terms, cardinal directions, and color terms. The ability of a language model to correctly ground conceptual terms is tested in a limited set of unseen environments and for unseen terms, where supervision is provided in the few-shot learning paradigm popularized with GPT-3. The largest tested model, GPT-3 175B, performs appreciably in all settings.

**Summary Of The Review:**

I think this is overall a nice paper with some nice ideas and interesting results, but it falls short of adequately addressing the research question that it is designed to. The biggest flaws in my view are the inclusion of too much supervision in the unseen-concept experiments, which make the results less relevant to the "isomorphism" question inasmuch as it is interesting to the community, and the exclusive reliance on the few-shot learning paradigm, which adds a big confounding factor to the results that makes it hard to interpret any clear trends. I do think that this is really close to being a very nice paper as long as these issues are adequately addressed at least in the framing of the work if not the experiments.

---

> ### Author Response · Authors · 2021-11-18
> **Thank you for your very detailed and very helpful review! We've answered and clarified several points in-line below.**
>
> “Deeper comparison and more credit to Abdou et. al”.
> We fully agree and have updated the related works section. We emphasise that while both papers look at the representations of colour terms in LMs, they are fundamentally different in that the Abdou et al paper looks at correlations between the linguistic space and grounded space. That paper never asks models to perform a mapping between the two spaces, which is the key focus in our paper. Thus, while they show that simple methods (PMI) might be sufficient to obtain a space i.e., reasonably well correlated with the grounded space, this is not contradictory with our results that show small models are unable to map the full linguistic space onto explicit representations of the grounded space.
>
> “Don’t think R-IV is more fair than R-ID”
> That is a fair point of disagreement, and we removed this claim, since this is a matter of opinion. We agree correctness vs. in-domain-ness are not the same, and we try to make clear this difference by providing multiple baselines and evaluation (e.g., we discuss results for in-domain-ness specifically and separately from correctness in section 3.3). R-IV/ID were intended purely as baselines, to help the reader understand how well a model could do if it only mastered in-domain-ness without making any headway on correctness.
>
> “GPT-2 models don't make sense of few-shot prompts in the effective way that GPT-3 does [..] separate issue from grounding.”
> This is an interesting and subtle point that you raise. We argue that in this task setting, few-shot learning _is_ an example of grounding, given how we set up the grounding task i.e., analogous to how most “understanding” or “reasoning” tasks are set up. For a model to succeed in this task, it must be able to ground terms and must be able to do it in a few-shot way. What you say is true--a model could fail because it is unable to learn in a few-shot manner and not because it is unable to ground (i.e., it might succeed at grounding if the task were set up differently). This is why we don’t make strong claims about the small models or that they are incapable of grounding, only that, based on our experimental results, large models _do_ succeed. Future work could no doubt refine these claims and present the small models in a more positive light by changing the task design. However, since there are so many ways one could frame the task, a thorough study of this warrants more than one paper. We try to keep our claims and conclusions modest and tied closely to the experiments we ran. Please let us know if we have over-stepped somewhere in our discussion of GPT-2’s capabilities, and we are happy to reword.
>
> “it limits the baselines that can be used [..] those from masked LMs..”
> We agree that testing different classes of models (e.g., masked LMs) is interesting, and we’ve added a set of experiments that does so in the Appendix in Tables 12 and 13, for a BERT-base model when given a similar input prompt. We outline the differences of this setting (from the previous generative autoregressive LMs) that might make the comparison somewhat unequal.
>
> “Including all n-1 concepts besides the target one [..]  provides a very strong constraint”
> To clarify:  we do *not* include all n-1 concepts for the colours, but rather include 60 out of 370 colours i.e., we test on the remaining 310 colours (not just one held-out colour). That said, we shared your concern about constraining the input, thus, we also include experiments that hold out entire colour sub-spaces (e.g., training only on 60 shades of red, rather than a random 60 colors; see Appendix B.1.1). The results remain well above random, indicating that the model is indeed generalizing well outside of training in a non-trivial way.
>
> “unseen-concept experiments need to more strongly limit their inputs”
> We report these results in Figure 6 in the Appendix on varying the number of samples in the input (from 5 up to 80 samples).
>
>
> “Rotations in RGB seems like it isn't the best way..”
> This is a very good point. We initially considered CIELAB instead of RGB, however, on viewing the Abdou et. al., dataset we realised that there was no ground truth colour for each CIELAB colour, but several human annotations about what they perceived the colour to be. This added an extra layer of complication to the evaluation, so we went with the RGB representations. We hope that, if anything, grounding to CIELAB should be easier than to RGB because CIELAB space is more perceptually meaningful.
>
> “Confusion about random world baseline”
> We apologise for the confusion. Your understanding is correct: the random world baseline does not preserve any meaningful distances between any two colours---they are a completely random distance, thus not preserving the structure of the space. Figure 2 might appear to have preserved some structure, but this is likely because it only shows 4 colors, so any orientation of the points will appear to have some structure to it.

---

> > ### Author Response · Authors · 2021-11-18
> > **[added additional points below]**
> >
> > “string matching Top-1 accuracy metric seems quite permissive”
> > We show examples of this metric in the Appendix C1. We agree that such metrics might be permissive, and like most string evaluation techniques, there are problems/edge-cases with all techniques. We include the distance analysis in Table 3 to assess how close the model’s generations are to the true label even when the string match fails. We also add an additional evaluation in the Appendix in Table 14, using exact-matches, which is a substantially less permissive metric. We see that even with this strict measure, the GPT-3 model matches the gold label a substantial portion of the time (36%) and that overall the results follow the same trends (i.e., GPT-3 model does substantially better than GPT-2 models, and is still above the baseline).
> >
> > “some confusion about the training data.”
> > Thank you for pointing this out! We apologise for the confusion about the training data and have updated all instances of this in the paper.
> >
> >  “including the requirement to generate an in-domain term”.
> > We don’t provide any such constraints. The baselines R-IV and R-ID are not learned models, but voca-based baselines e.g., selecting one out of all vocabulary words (R-IV) or one out of all in-domain words (R-ID).

---

> > > ### Comment · Reviewer_qf1D · 2021-11-23
> > > **Thanks for your responses!**
> > >
> > > Sounds good! Just following up on a few points below.
> > >
> > > On **testing grounding with few-shot learning:** I think the caveat is important to state upfront. When people see and read the results tables, they will think *more parameters means grounding better.* Of course, this is by all accounts true, but how accurately do the results represent this trend? For example, Appendix D says:
> > > > More concretely, the generalisation extent of each model seems to max out at some number of input prompts, with a trend of increasing performance on increasing model size. Therefore, a larger model (than the ones we have here) might be required in order to achieve better grounded generalisation.
> > >
> > > This directly ties the trends in few-shot learning results to grounded generalization capabilities. This is problematic to state without the extra assumption that grounding has to happen via few-shot learning. The paper also still frames itself as directly addressing the question of an "isomorphism" of meaning spaces (Introduction):
> > > >  In principle, it is therefore possible for a text-only model’s conceptual space to be isomorphic to the “true”, grounded conceptual space (Merrill et al., 2021). In this work, we investigate whether this is the case by asking whether models can learn to ground an entire domain (e.g., direction) after grounding only a subset of the points in that domain (e.g., left).
> > >
> > > This seems to suggest that going from any subset to the full set is sufficient to test what is meant under the "isomorphism" criterion. The actual situation is more complicated than this as I described in my review. Going from n-1 to n requires very little in the way of an "isomorphic" meaning space: indeed, it can be solved entirely if you 1) know the set of in-domain terms, and 2) follow pragmatic assumptions of mutual exclusivity of reference. And you may not have done the full n-1 for color, but the same factors still apply to some extent for large n. The input overconstrains the meaning space, so it could be helping the model correct its internal "errors" relative to the ground-truth space. Anyway, I just think some of the implications in the framing here should be weakened if the more complicated issues aren't going to be discussed in the paper.
> > >
> > > Regarding **the constraint to generate an in-domain term,** what I meant is I think it would be a good ablation test to see how the GPT models do under that constraint; I suspect it would noticeably narrow the gap between the small and large models, and do a lot to address my complaint about the smaller models not knowing how to do few-shot learning.
> > >
> > > Regarding **rotations in CIELAB space**: what I meant was you could translate the RGB color to CIELAB, execute a rotation, and then translate it back to RGB. This should preserve perceptual distance while still giving you RGB colors. It's not a big deal though.

---

### Official Review · Reviewer_TAjy · 2021-11-03

**Correctness:** 4
**Technical Novelty And Significance:** 3
**Empirical Novelty And Significance:** 3
**Recommendation:** 8
**Confidence:** 4

**Details Of Ethics Concerns:**

I have no ethics concerns for this paper.

**Main Review:**


Overall I really enjoyed this paper.  The scope was fairly narrow and the experiments were not diverse enough to provide a sense of a thorough understanding of how LM's are forming this space, what possible confounds might be at play, or what other domains might have similar effects.  But apart from these concerns, the paper is well-written and presented in an organized manner, and essentially all the discussion points contributed something to my understanding of the work.  The topic is also very timely and important, and the implications of this work could help unify work in NLP, vision, and other grounded/situated learning.

An obvious main strength of the paper is that empirically the results are clear and consistent across all experiments.  The similarities between induced and gold spaces, between smaller models and larger models, is very significant.  The experiments themselves are quite simple and straightforward, but my biggest worry -- that confounds from the LM training data may be leaking in and influencing these conceptual spaces -- were strongly considered by the authors.  Their solution of using rotations to test against isomorphic spaces makes intuitive sense.  One thing I am considering is whether it's possible that huge parameter language models may be inducing mechanisms for performing spatial rotations on subspaces, though at some point it might become difficult to separate a model's ability to memorize the data and perform on-the-fly rotations, from something that might be called a non-trivial understanding of these concepts/spaces.  That is mostly an aside as I can't present it as anything more than speculation.

One of the few questions about the color experiment results was to what extent the model was backing off to predict less specific color names, since it is something that is not well-captured in top-1 accuracy (though maybe some combination of this and distance evaluation starts to get at it).

And why were 67 other colors inserted in the color prompts?  It seemed arbitrary.

How are grids incorporated into the model?  In Appendix B2, it appears to be just as a string following "World", but is it a "linearized" string of one row after another?


Typos:
Zellers reference
"within the prompt ."




**Summary Of The Paper:**

This work seeks to probe large pre-trained language models for indications that they have induced a conceptual space that is structured similarly to one constructed from interactions with the real-world, despite being trained solely from text.  Most of this probing takes the form perceptual tasks -- can LMs quickly learn new concepts related to navigation, or colors, if given some related examples from the space?  In both navigation and color tasks, this seems to be the case, and a consistent trend emerges where larger LMs perform better on this task.  The experiments seem well-controlled, and the authors present an insightful discussion of model performance.



**Summary Of The Review:**

The paper provides new insights into whether large pre-trained LMs induce conceptual spaces that mirror those found in the real world.  This could be an important finding, that together with many related works, gives us a better understanding of what sorts of higher reasoning abilities such LMs are capable of.  I found the experiments straightforward, but well-done, and some foresight was given to what confounding factors may be creeping into these results.  And across all model sizes we learn that similarities between these spaces increases with model size, and that low model sizes, though still big in comparison to many models, here are truly insufficient to capture concept spaces anywhere comparable to the largest models.  The paper is well-written and references are good, although I may have missed an important reference or two since this is slightly outside of my research area, but barring the existence of similar uncited work, I think it's a paper worth publishing.

---

> ### Author Response · Authors · 2021-11-18
> **Thank you for your review. We have answered questions in-line below.**
>
> “And why were 67 other colors inserted in the color prompts? It seemed arbitrary.”
> The number of samples we include in a prompt is dependent on the context width of the models (e.g., 1024 characters) as well as the size of each sample in a domain (e.g, large arrays for the spatial domain take up more characters than an RGB code in the colour domain). For each experiment, we attempt to maximise the number of samples we can give to the model (that fit in its context-width) and use these in the prompt. This amounts to 67 for the colour domain, and around 20 for the spatial and cardinal domains.
>
> “How are grids incorporated into the model? In Appendix B2, it appears to be just as a string following "World", but is it a "linearized" string of one row after another?”
> This is correct, each grid-world array is a linearised string, where each row is separated by a “\n” character. Although this representation has imperfections, we find that models can, to some degree, learn spatial/cardinal directions using this.
>
> “One of the few questions about the color experiment results was to what extent the model was backing off to predict less specific color names, since it is something that is not well-captured in top-1 accuracy (though maybe some combination of this and distance evaluation starts to get at it).”
> Especially for the GPT-3 models, we see that models do seem to have a tendency to predict less specific colour names. Especially seeing that the difference between accuracy and the distance metrics, it seems like models tend to predict colours that are close in colour space, however the colour name differs and is often less specific (e.g., saying “red” instead of “dark red”). To assess this better, we also add an additional evaluation with exact-matches in the Appendix in Table 14, which is a substantially less permissive metric. We see that even with this strict measure, the GPT-3 model matches the gold label a substantial portion of the time (36%) and that overall the results follow the same trends (i.e., GPT-3 model does substantially better than GPT-2 models, and is still above the baseline).

---

### Official Review · Reviewer_BFAr · 2021-11-03

**Correctness:** 3
**Technical Novelty And Significance:** 3
**Empirical Novelty And Significance:** 3
**Recommendation:** 8
**Confidence:** 4

**Main Review:**

I think the modelling work presented is useful and interesting — and importantly: explained well and coherently. However, I find many statements not thought out as much as they deserve. For one, colours are as relative as left/right. Something red looks dramatically different under white light than it does under other colours of light, and so on. Also see visual illusions where the same shade of grey looks like different colours as a function of where it is in the image. This is why categorical cognition plays a role in colours in humans. The authors could at minimum note this, if they wish. Another example is on the equivocation of "text" and "language" throughout the manuscript. I also find it a bit strange to say "north" in humans is not grounded. I assume it is always grounded to where the Sun is and what hemisphere we are in and so if somebody were to lie to us about where north is, we would eventually figure it out when we saw the movement of shadows, etc. There is a lot of research on what humans indeed do here, so some literature review could be done if claims need to be made about actual human behaviour.

Overall, however, I find the modelling here very interesting and compelling. I like that the authors explored a pre-trained model. Perhaps providing a coherent definition of "grounded" versus "ungrounded" might be useful to help formalise the distinction as used by the authors.

**Summary Of The Paper:**

The authors present their work looking at how text-only input can create what they call a "grounded" model. The results support the idea that a small input of text-only data can give rise to the models behaving in such ways as to generalise over things like left and right or colours. This indicates, the perhaps unsurprising conclusion, that language is able to, on a meta level, encode some aspects of groundedness, as defined by the authors, and the perhaps more impressive conclusion that the models can indeed tap into the latent structure found in language using a few examples and thereby generate useful isomophisms.

**Summary Of The Review:**

The work is good quality and well-explained, but could benefit from slightly more details on the rationale behind some jargon and claims about human cognition.

---

> ### Author Response · Authors · 2021-11-18
> **Thank you for your review. We've addressed concerns in-line below.**
>
> “Another example is on the equivocation of "text" and "language" throughout the manuscript.”
> This is a really good point. The reason there is an equivocation of “text” and “language” is because of the current regimen in NLP that “learns language” by inputting text into models---but we agree with your point that this should not be the case and that language encompasses far more than just text.
>
> “ I also find it a bit strange to say "north" in humans is not grounded.”
> We agree and will downplay the human comparisons. We used the notion of being “disoriented” only to provide intuition, not to make strong claims about how humans work. It is true that north should always be grounded, but for an individual human, if they happened to think “south” is “north”, they would then map the others (e.g., “east” and “west”) accordingly.

---

### Official Review · Reviewer_e6Xz · 2021-11-05

**Correctness:** 2
**Technical Novelty And Significance:** 3
**Empirical Novelty And Significance:** 3
**Recommendation:** 6
**Confidence:** 4

**Main Review:**

I think this paper investigates a very interesting problem. The experiments are rather thorough, with different levels of controls (e.g., semantic-invariant transformations to the world representations, generalization to unseen worlds or unseen concepts). The writing is clear and structured as well.

However, I think the concerns that some readers might raise and complain include:

(1) Metric. Is the top-3 accuracy meaningful for the task, especially for the spatial and cardinal problems where the concept space is very small, and for GPT-3 that knows to output in-domain words only? Is the substring metric suitable for the color problem, especially in the “unseen concept” setup? For example, if “light blue” is a seen concept and “dark blue” is the test-time unseen concept, then answering the *seen* concept “light blue” in the unseen concept setup would result in a perfect accuracy. Would that defeat the purpose of testing generalization to unseen concepts, like in Table 2?

(2) Conclusion drawn from the results. The authors argue that if the LM successfully generates the correct concept based on the grounded representation (likely “unseen” in the pretraining data), it means that the model knows to ground the concept to the non-text world. However, is it possible that the model doesn’t understand the relationship between the concepts and the grounded representations, but instead utilizes a similarity between the test grounded representation and the grounded representations in the in-context prompts? For example, upon seeing the test representation (e.g., [0,1,0,0] in the spatial domain, or RGB (140, 0, 255) in the color domain), the model can use a simple strategy: copying the concept of a bunch of most similar representations in the in-line prompt examples (e.g., [0, 1, 0, 0, 0], or RGB (145, 0, 255)). This strategy would not involve the concept of “left” or “pink”, and is robust to the rotation transformation (while not robust to the random transformation, if each point in the world was transformed independently). This would align with the results in Table 1. To check whether this hypothesis is (partially) true, we can look at experiments like:

A: Test on some real unseen concepts. This was done in the paper, like in spatial and cardinal columns in Table 2, Table 9, 10, 11 (in the appendix). But the performance is not very strong in these cases even for GPT-3 (top-1 accuracy).

B: Test with fewer prompts. This is to prevent the model from memorizing similarity with the prompting examples too much. This was also done in the paper (Figure 6 in the appendix). Again, the performance is not strong, if the number of prompts goes below 20 or 60.

C: Replace all of the concept names with concepts in an unrelated domain (e.g., substituting all “left” and “right” with “apple” and “orange”). If the performance is above baseline in this setup, should we conclude that the LM implicitly learns to map fruit concepts to the grounded spatial world? This was not done in the paper and may be a good control experiment.

(3) Though the paper investigates an interesting problem, the overall takeaway of this work is not very clear to me. How is the analysis useful for future work?

(4) Some details in the paper should be checked. For example, in Section 2.1, the authors say that all models are pretrained on the 40GB OPENAI-WT dataset, but this is not true for GPT-3? Also for the color experiments, it is not clear whether 60 or 6+57 or 70 (as mentioned in B.1.1 in the appendix) prompts were used.


**Summary Of The Paper:**

This works aims at investigating whether large language models (LMs) pretrained only on texts, can implicitly learn grounded concepts of the world beyond texts. Specifically, the authors test whether the LMs can map some conceptual domains (e.g., direction, color) to grounded world representations (e.g., textualized grid world, RGB representation of colors). The authors give a rather small number of examples as prompts to the models in an in-context learning setup. They find that large LMs like GPT-3 can often output the correct concept for the grounded world representations, even though it’s likely that the model hasn’t seen the grounded world representations in its pretraining. They conclude that the text-only LMs may already learn the grounded representations implicitly, without explicit from-scratch training like in the visual language models.


**Summary Of The Review:**

Overall I think this work investigates an interesting problem, but the main argument needs to be justified more carefully (as mentioned in (1), (2) above). Also, the takeaway and impact of the work are not very clear to me, other than showing the somewhat inscrutable power of GPT-3.

---

> ### Author Response · Authors · 2021-11-18
> **Thank you for your review. We address specific points in-line below.**
>
> “is the substring metric suitable for the color problem especially in the unseen concept setup?”
> We agree with the reviewer’s concerns about metrics and acknowledge that evaluating model generations (when they are n-length sequences)  has been an open problem in NLP. The substring metric that we use evaluates whether the model-generated answer is a substring on the true answer, thus favouring brevity and shorter/more pragmatic responses (e.g., “blue” would be correct for “dark blue”, but “light blue” would not be correct for “dark blue” or even “blue”). To alleviate this concern, we evaluate exact-match string accuracy (which is a substantially less permissive metric than substring-match) in Table 14 in the Appendix. We see that even with this strict measure, the GPT-3 model matches the gold label a substantial portion of the time (36%) and that overall the results follow the same trends (i.e., GPT-3 model does substantially better than GPT-2 models, and is still above the baseline).
>
> “possible that the model doesn’t understand the relationship between the concepts and the grounded representations, but instead utilizes a similarity between the test grounded representation and the grounded representations in the in-context prompts”
> We argue that this similarity, given the nature of the input prompts, does imply grounding. Specifically for the domains we test, the “similarity” between two instances e.g., two grid-worlds, depends on the relative distance in space between the objects of interest.
>
> “the overall takeaway of this work is not very clear to me. How is the analysis useful for future work?”
> Although the work in this paper focuses on only a small set of grounded concepts (colours, spatial terms, cardinal directions) when represented in simple grid-worlds, the main takeaway from this work is that it *is* possible to take a text-only language model and teach it some grounded concepts. We acknowledge that we are limited in the types of concepts we can represent (e.g,. sensory input is far harder to represent in a grid-world), but we think that the findings from this work demonstrate how we could transfer a language model trained only on text onto a non-textual domain. This is exciting because text data is much cheaper and easier to obtain than multimodal data. If it is possible to learn a conceptual space from text, and then map it to a different perceptual space with comparatively few examples, that implies a much more efficient way of obtaining multimodal models. This is a long-term research direction that will require much more research, but this paper shows its promising and exciting to pursue.
>
> “Some details in the paper should be checked. For example, in Section 2.1, the authors say that all models are pretrained on the 40GB OPENAI-WT dataset, but this is not true for GPT-3? Also for the color experiments, it is not clear whether 60 or 6+57 or 70 (as mentioned in B.1.1 in the appendix) prompts were used.”
> We apologise for the confusion and have updated the paper to clarify the pretraining data and number of prompts used.

---

> > ### Comment · Reviewer_e6Xz · 2021-11-30
> > **Thanks for your responses**
> >
> > Just following up on the second point, regarding “we argue that this similarity, given the nature of the input prompts, does imply grounding”: I agree this “similarity” could imply “grounding” in general, but my point was more about whether this shows language grounding over the specific investigated concepts (e.g., left or right). In point 2-C of my original review, I mentioned an imaginary scenario of swapping investigated concepts like left or right, with irrelevant control concepts like apple or orange. If these irrelevant/control fruit concepts also work in the grid-world setup, then it may suggest that the model is aware of the grid-world spatial similarity, but *not* that the investigated direction terms are grounded to this spatial awareness. Overall, it might be good to consider adding a short discussion on this point, since it’s a hypothesis of *why* GPT-3 works in your problem. Throughout the work, a lot of efforts are spent on showing the fact that GPT-3 works, but probably not discussing why GPT-3 works (even as conjectures).

---

### Public Comment · ~Jason_Wei1 · 2022-02-14
**I found this paper very interesting**

I try to avoid reading papers whenever possible (I know, controversial take), but this one was so interesting I had to write up my thoughts. I’m probably misunderstanding some parts, so please correct me if I’m wrong.

Overall this is a high-quality paper in many regards: motivation, thoroughness, clarity, clever isophomic space setup, but you already knew that from the reviews.

As context, this is the first paper I have read in the grounded language space, so I am likely missing stuff.

OK, so the main perplex of this paper, if I’m understanding correctly, is that you must draw one of the two conclusions.

- GPT-3 can ground concepts, which implies that true meaning *can* be learned from form, which means this paper refutes Bender and Koller, right?
- The model’s non-random performance comes from some spurious correlations that were missed in the experimental setup.

Am I wrong here? Obviously you can’t say Conclusion 2 in a conference submission, but you hesitated to say Conclusion 1 forcefully (maybe because it’s hard to be sure it’s not 2).

Here’s some things that I think may decrease the accuracy (after all, we have to say conclusion 1 or conclusion 2, and conclusion 2 seems much more likely).

- The “apple” and “orange” experiment suggested by reviewer e5Xz seems worthwhile.
- The hold-out splits can be potentially problematic. If you prompt with east, west, north, and test on south, couldn’t the model know to predict south just by the fact that the test-time example was different from the few-shot exemplars? Can you use {north, south} in the exemplars and test on {east, west}?
- The isomorphism as a control is good, but I think it’s possible that rotated worlds appeared in the training data. Just to demonstrate how bad intuition is, how many times do you think the word “amalthough” appeared in the training data of C4? (Google will spellcorrect you if you google it.) Well, it appeared 12,093 times. So I think it’s safe to say that a rotated world probably appeared in the training set. - - Change the inputs to made-up words (e.g., make all zeros “semicephaly” and ones “atrocitious”) and I will be surprised if you get above-chance accuracy (though I’m sure you can come up with reasons why this is too adversarial).

The other challenging thing about this paper is that the numbers are like not that good, but they’re definitely not random. It’s hard to make any strong statements about the numbers. For instance, .45 on a 6-way classification.
- Side note, what’s the difference between above and up? If you show the model above, then I no longer consider “up” to be out of distribution in the sense that it can know what “up” means just by knowing that it’s a synonym of above. This seems crucial when your accuracy is .45. Even if you accounted for it in the eval, why not just have four directions?
- Another side note, it would have been better to restrict the model’s outputs by simplifying scoring the possible continuations and comparing their scores. Then you don’t have to dilute your paper with stuff like “the smallest models often produce random, unrelated outputs”
- Another side note, I feel like top-3 accuracy was not worth the space it took up

Wasn’t clear to me how the “random” setting works in the spatial/direction tasks, but I confess I you probably explained it and i missed it.

---

### Decision · Program_Chairs · 2022-01-20

**Decision:**

Accept (Poster)

**Comment:**

The authors explore the hypothesis of whether grounded representations can be leaved from text only. They show that a language model trained with relatively little data can make conceptual domains such as color to a grounded world representation such as RGB coordinates. The paper was positively received by the reviewers, specifically after a fruitful discussion to further clarify the points that the authors were making and their conclusions. The authors have already edited some parts of the paper, I ask them to go back and include other points that the reviewers have made. I recommend this paper for acceptance, it will generate good discussion and ideas at ICLR.